



# Evaluation of the absorption Ångström exponents for traffic and wood burning in the Aethalometer based source apportionment using radiocarbon measurements of ambient aerosol

Peter Zotter[1,a], Hanna Herich[2], Martin Gysel[1], Imad El-Haddad[1], Yanlin Zhang[1,3,4,5,b], Griša Močnik[6,7], Christoph Hüglin[2], Urs Baltensperger[1], Sönke Szidat[3,4], André S.H. Prévôt[1]

[1]Laboratory of Atmospheric Chemistry, Paul Scherrer Institute (PSI), 5232 Villigen PSI, Switzerland
[2]Laboratory for Air Pollution and Environmental Technology, Swiss Federal Laboratories for Materials Science and Technology (Empa), Überlandstrasse 129, 8600 Dübendorf, Switzerland
[3]Department of Chemistry and Biochemistry, University of Bern, Bern, Switzerland
[4]Oeschger Centre for Climate Change Research, University of Bern, Bern, Switzerland
[5]Laboratory of Radiochemistry and Environmental Chemistry, Paul Scherrer Institute (PSI), 5232 Villigen PSI, Switzerland
[6]Aerosol d.o.o., Research and Development Department., Ljubljana, Slovenia
[7]Condensed Matter Physics Department, Jožef Stefan Institute, Ljubljana, Slovenia
[a]now at: Lucerne University of Applied Sciences and Arts, School of Engineering and Architecture, Bioenergy Research, Technikumstrasse 21, CH-6048 Horw, Switzerland
[b]now at: Yale-NUIST Center on Atmospheric Environment, Nanjing University of Information Science and Technology, 210044, Nanjing, China

*Correspondence to:* A.S.H. Prévôt (andre.prevot@psi.ch)

**Abstract.** Black carbon (BC) measured by a multi-wavelength Aethalometer can be apportioned to traffic and wood burning. The method is based on the differences in the dependence of aerosol absorption on the wavelength of light used to investigate the sample, parameterized by the source-specific Ångström absorption exponent ($\alpha$). While the spectral dependence (defined as $\alpha$ values) of the traffic-related BC light absorption is low, wood smoke particles feature enhanced light absorption in the blue and near ultraviolet. Source apportionment results using this methodology are hence strongly dependent on the $\alpha$ values assumed for both types of emissions: traffic $\alpha_{TR}$, and wood burning $\alpha_{WB}$. Most studies use a single $\alpha_{TR}$ and $\alpha_{WB}$ pair in the Aethalometer model, derived from previous work. However, an accurate determination of the source specific $\alpha$ values is currently lacking and in some recent publications the applicability of the Aethalometer model was questioned.

Here we present an indirect methodology for the determination of $\alpha_{WB}$ and $\alpha_{TR}$ by comparing the source apportionment of BC using the Aethalometer model with [14]C measurements of the EC fraction on 16 to 40 h filter samples from several locations and campaigns across Switzerland during 2005-2012, mainly in winter. The data obtained at eight stations with different source characteristics also enabled the evaluation of the performance and the uncertainties of the Aethalometer model in different environments. The best combination of $\alpha_{TR}$ and $\alpha_{WB}$ (0.9 and 1.68, respectively) was obtained by fitting the Aethalometer model outputs (calculated with the absorption coefficients at 470 nm and 950 nm) against the fossil fraction of EC ($EC_F/EC$) derived from [14]C measurements. Aethalometer and [14]C source apportionment results are well correlated ($r = 0.81$) and the fitting residuals exhibit only a minor positive bias of 1.6% and an average precision of 9.3%. This indicates that the Aethalometer model reproduces reasonably well the [14]C results for all stations investigated in this study using our best estimate of a single $\alpha_{WB}$ and $\alpha_{TR}$ pair. Combining the EC, [14]C and Aethalometer measurements further allowed assessing the dependence of the mass absorption cross section (MAC) of BC on its source. Results indicate no significant difference in MAC at 880nm between BC originating from traffic or wood burning emissions. Using $EC_F/EC$ as reference and constant a priori selected $\alpha_{TR}$ values, $\alpha_{WB}$ was also calculated for each individual data point. No clear station-to-station or season-to-season differences in $\alpha_{WB}$ were observed, but $\alpha_{TR}$ and $\alpha_{WB}$ values are interdependent. For example, an increase in $\alpha_{TR}$ by 0.1 results in a decrease in $\alpha_{WB}$ by 0.1. The fitting residuals of different $\alpha_{TR}$ and $\alpha_{WB}$ combinations depend on $EC_F/EC$ such that a good agreement cannot be obtained over the entire $EC_F/EC$ range using other $\alpha$ pairs. Additional combinations of $\alpha_{TR} = 0.8$, and 1.0 and $\alpha_{WB} = 1.8$ and 1.6, respectively, are possible but only for $EC_F/EC$ between ~40% and





85%. Applying $\alpha$ values previously used in literature such as $\alpha_{WB}$ of ~2 or any $\alpha_{WB}$ in combination with $\alpha_{TR}$ = 1.1 to our data set results in large residuals. Therefore we recommend to use the best $\alpha$ combination as obtained here ($\alpha_{TR}$ = 0.9 and $\alpha_{WB}$ = 1.68) in future studies when no or only limited additional information like $^{14}$C measurements are available. However, these results were obtained for locations impacted by BC mainly from traffic consisting of a modern car fleet and residential

wood combustion with well-constrained combustion efficiencies. For regions of the world with different combustion conditions, additional BC sources or fuels used further investigations are needed.

## 1 Introduction

Recently, the World Health Organization (WHO) reported around 3.7 million premature deaths in 2012 as a result of exposure to ambient air pollution, demonstrating that health risks in areas of low air quality are far greater than previously thought (WHO, 2014). Atmospheric particulate matter (PM) contributes significantly to ambient air pollution and adversely affects human health causing respiratory and cardiopulmonary diseases associated with increased morbidity and mortality (Pope and Dockery, 2006; WHO, 2006). Although PM levels were decreasing in the last decade in Europe and also in Switzerland, legal thresholds are still exceeded (Barmpadimos et al., 2011; 2012). Carbonaceous material (total carbon, TC) is a major fraction of the fine aerosol mass (up to 90% of the PM mass < 2.5 μm, Gelencsér, 2004; Putaud et al., 2004; Jimenez et al., 2009) and is further classified into the sub-fractions organic carbon (OC) and black carbon (BC) or elemental carbon (EC) (Jacobson et al., 2000). BC is the light-absorbing part of carbonaceous material and, compared to other aerosol components, it contributes significantly to global warming due to its optical and radiative properties (Jacobson, 2001 and 2010; IPCC, 2013). Because of the relatively short atmospheric lifetime of BC, its radiative forcing ends within weeks after emission. Thus reducing BC emissions may rapidly reduce climate warming (Shindell et al., 2012; Bond et al., 2013 and references therein). Therefore, the identification of different BC sources and their emission strength is crucial for the implementation of effective mitigation strategies.

The emission sources of BC are combustion processes of fossil and non-fossil carbonaceous fuels. In Switzerland, large parts of Europe and other parts of the world, BC mainly originates either from traffic or biomass burning in winter (e.g. Szidat et

al., 2007; Favez et al., 2010; Lanz et al., 2010; Piazzalunga et al., 2011; Harrison et al., 2012; Larsen et al., 2012; Crippa et al., 2013; Herich et al., 2014). Different methods exist to quantify carbonaceous aerosol fractions based on light absorption, thermo-optical or laser-induced incandescence measurements. The quantities measured are defined based on the instrument and protocol used, with BC and EC related to light absorption and thermo-optical measurements, respectively (Petzold et al., 2013). In recent years, the Aethalometer, an online measurement technique of the aerosol light absorption at seven different

wavelengths ranging from near-ultraviolet (N-UV) to near-infrared (N-IR), has become widely used, since it is rather inexpensive, portable, easy to operate and suitable for long-term measurements. Furthermore, multi-wavelength Aethalometer data may be used to derive the traffic and the wood burning contributions to BC (BC$_{TR}$ and BC$_{WB}$, respectively) taking advantage of the light absorption in the blue and N-UV of aerosols from biomass combustion likely due to co-emitted organics, which is enhanced compared to aerosols from fossil sources (Sandradewi et al., 2008a). The so-called

"Aethalometer model" assumes that light absorbing particles only originate from vehicle and biomass burning emissions, and uses absorption Ångström exponent ($\alpha$) values specific to these sources to derive their contributions. Therefore, the source apportionment of BC using the Aethalometer model is inherently dependent on the a priori assumed absorption Ångström exponents for traffic ($\alpha_{TR}$) and biomass burning ($\alpha_{WB}$), which are based on a few emission studies. $\alpha_{TR}$ values cluster in a narrow range (0.8–1.1), whereas a large range of $\alpha_{WB}$ values (0.9 to 3.5) is reported (Schnaiter et al., 2003;

Kirchstetter et al., 2004; Schnaiter et al., 2005; Lewis et al., 2008; Saleh et al., 2013). Some studies also obtained $\alpha_{TR}$ from ambient Aethalometer measurements by investigating the $\alpha$ values calculated from the ambient absorption coefficient ($b_{abs}$) values of the total light absorbing aerosol during periods and locations that were only influenced by traffic emissions (e.g. in





summer close to roads, Sandradewi et al., 2008b, Herich et al., 2011). It should be noted, however, that $\alpha$ values depend not only on different emission sources but also on the choice of wavelengths and different calculation procedures used, although deviations due to the latter are expected to be low (Moosmüller et al., 2011).

Another independent and more direct approach than the Aethalometer model to distinguish between modern (wood burning)
and fossil (traffic) contributions is the radiocarbon analysis. Radiocarbon ($^{14}$C) is completely depleted in fossil fuel emissions ($^{14}$C half-life = 5730 years) and can, therefore, be separated from non-fossil carbon sources, which have a similar $^{14}$C signal as atmospheric carbon dioxide ($CO_2$) (Currie, 2000; Szidat, 2009). Measuring $^{14}$C in the EC fraction therefore directly enables the quantification of the wood-burning and fossil sources of EC. However, the $^{14}$C analysis can only be performed on filter samples and is therefore limited in time resolution. Furthermore, such analysis is rather expensive and time consuming.
The $^{14}$C measurement in the EC fraction remains additionally challenging in contrast to TC (Szidat et al., 2013), since a clear physical separation between OC and EC is necessary to avoid interferences from OC in the $^{14}$C signal. Nevertheless, recent developments and method adaptations from different groups show more consistent approaches and yield more robust $^{14}$C results (Zhang et al., 2012; Bernardoni et al., 2013; Dusek et al., 2014).

Sandradewi et al. (2008a) first employed the Aethalometer model on winter data from a polluted Swiss alpine valley and
used $^{14}$C measurements of the EC fraction to test the assumed $\alpha_{WB}$ and $\alpha_{TR}$. Based on this work, subsequent studies using the Aethalometer model employed similar $\alpha_{TR}$ (0.9–1.1) and $\alpha_{WB}$ (1.8–2.2) values (e.g. Sandradewi et al., 2008b; Favez et al., 2010; Perron et al., 2010; Herich et al., 2011; Harrison et al., 2012; Crippa et al., 2013; Harrison et al., 2013; Mohr et al., 2013), without further evaluations of these parameters using external data. Others compared the Aethalometer model outputs to apportionments using specific source tracers (Favez et al., 2010; Herich et al., 2014; Crilley et al., 2015). However, such
approaches heavily rely on a-priori assumed tracer-to-BC emission ratios, which are highly variable (Schmidl et al., 2008; El Haddad et al., 2011; Heringa et al., 2011; El Haddad et al., 2013), and as such are not suitable for the evaluation of the $\alpha_{TR}$ and $\alpha_{WB}$ values used in the Aethalometer model. Even though the Aethalometer model is widely used there are also studies (Harrison et al., 2013, Garg et al., 2016) that question the applicability of this model when other and/or additional combustion sources may contribute to the BC burden and combustion efficiencies are less well constrained.

In this study we present an evaluation of the Aethalometer model by comparing its outputs to $^{14}$C results of the EC fraction in order to validate the choice of the Ångström exponents for wood burning ($\alpha_{WB}$) and traffic emissions ($\alpha_{TR}$). To this end, we use $^{14}$C and Aethalometer data from different campaigns across Switzerland, mostly from the winter season. The dataset in this study (n = 101) is significantly larger than previously reported (n = 12 and n = 18 in Sandradewi et al. (2008a) and (2008b), respectively). In addition, the data presented here were obtained at eight different stations in various area types with
different source characteristics (e.g. urban, suburban, rural, alpine valley, traffic, background, etc.) thereby enabling the evaluation of the performance and the uncertainties of the Aethalometer model in different environments.

## 2. Materials and methods

### 2.1 Aerosol sampling

Aerosol sampling presented in this study (see Table 1) was carried out at several stations of the Swiss National (NABEL)
and Cantonal air pollution monitoring networks (EMPA, 2013; Cercl'Air, 2012). The stations ZUR, PAY, REI and SIS are located north of the Alps, whereas MAG, ROV and MOL are located south of the Alps and MAS is situated in the Rhone valley. The location of these stations in different areas allowed the sampling of a broad range of particles, with different characteristics ranging from urban to rural and from traffic to background. The exact locations of the stations are shown in Fig. 1 and the details and full names of the sites as well as the different campaigns carried out at these stations are listed in
Table 1.




Filter sampling was conducted using quartz fiber filters (Pallflex 2500QAT-UP) and high-volume samplers (Digitel DHA-80, Switzerland) at a flow rate of 30 m$^3$ h$^{-1}$. The collection time as well as the size cut of the PM inlet varied between 16 h and 40 h as well as PM$_1$ and PM$_{10}$, respectively, depending on the campaign (see Table 1). After sampling, filters were stored at -20 °C until analysis. Most of the results presented here (n = 69) were obtained on PM$_{10}$ filters with a sampling time

of 24 h from the five-year $^{14}$C project Switzerland (Zotter et al., 2014). The samples from this campaign were collected on days with high PM$_{10}$ concentrations (almost all of them exceeding the Swiss and EU daily limit of 50 µg m$^{-3}$). The period covers mainly the winter season at SIS, PAY, MAG and ZUR and only few samples from spring and summer at ZUR were analyzed. Filter samples from earlier studies (n = 32) across Switzerland in 2005 at MOL, REI, MAS and ROV as well as in 2006 at ZUR were only collected in winter during shorter campaigns (~1 month).

BC has been continuously measured at the NABEL stations MAG (since 2008), PAY (since 2008) and ZUR (since 2009) using a 7-wavelength Aethalometer (MAGEE Scientific, model AE31) (Herich et al., 2011; EMPA, 2013). The same type of instrument was also placed at SIS in the winters 2010/2011 and 2011/2012 during the $^{14}$C project Switzerland and earlier campaigns in 2005 and 2006. In total 101 samples with parallel $^{14}$C and Aethalometer measurements are available (n = 9, 24, 19, 19, 13, 4, 5, 8 for SIS, ZUR, MAG, PAY, ROV, MOL, REI and MAS, respectively).

## 2.2 Aethalometer

### 2.2.1 Measurement Principle

The Aethalometer provides a real-time optical measurement of light absorbing carbonaceous aerosols at seven wavelengths ($\lambda$ = 370, 470, 520, 590, 660, 880 and 950 nm; Hansen et al., 1984, Hansen, 2003). It measures the attenuation (ATN) of a light beam transmitted through a filter on which aerosols are continuously collected.

$$ATN = 100 \cdot \ln\left(\frac{I_0}{I}\right) \tag{1}$$

where I$_0$ and I denote the intensity of a light beam through an empty and particle-laden spot of a filter tape, respectively. The change in ATN over a certain time period ($t$) is proportional to the attenuation coefficient ($b_{ATN}$) given a known flow rate ($Q$) and spot size ($A$) onto which particles are collected.

$$b_{ATN} = \frac{A}{Q} \cdot \frac{\Delta ATN}{\Delta t} \tag{2}$$

Like all filter-based absorption techniques, the Aethalometer uses integration of the sample on the filter to increase the sensitivity of the measurement. Scattering by the filter fibres enhances absorption of the light by the aerosols collected on the filter tape. As the filter gets loaded by light absorbing aerosols and ATN increases, non-linear loading effects become apparent (Liousse et al., 1993; Petzold et al., 1997; Bond et al., 1999; Park et al., 2010; Drinovec et al., 2015). To compensate for these effects, the algorithm developed by Weingartner et al. (2003) was used to derive the final absorption

coefficient ($b_{abs}$)

$$b_{abs}(\lambda) = \frac{b_{ATN}(\lambda)}{C_\lambda \cdot R(f_\lambda, ATN_\lambda)} \tag{3}$$

where $C_\lambda$ and $R(f_\lambda, ATN_\lambda)$ are factors to compensate for multiple scattering of the filter fibres and the loading effect, respectively.

$$R(f_\lambda, ATN_\lambda) = \left(\frac{1}{f_\lambda} - 1\right) \cdot \frac{\ln(ATN_\lambda) - \ln(10)}{\ln(50) - \ln(10)} + 1 \tag{4}$$

In Eq. (4) $f_\lambda$ denotes the slope between the linear function $R(f_\lambda, ATN_\lambda)$ vs. $\ln(ATN_\lambda)$ and allows estimating the instrumental error that occurs when the shadowing effect is disregarded (Weingartner et al., 2003). This approach is routinely applied to the Aethalometer data from the NABEL stations using a single $C$ value of 2.14 for all wavelengths and wavelength



dependent $f$ values (1.155, 1.137, 1.128, 1.116, 1.103, 1.064 and 1.051 for 370 nm, 470 nm, 520 nm, 590 nm, 660 nm, 880 nm and 950 nm, respectively) as proposed by Weingartner et al. (2003) and Sandradewi et al. (2008c), respectively. The same values were also used to compensate the data from SIS and the previous campaigns in Switzerland. Several other algorithms for the compensation of the Aethalometer data are available (Collaud Coen et al., 2010 and references therein)

and some studies slightly adapted the Weingartner et al. (2003) approach (Sandradewi et al., 2008c; Favez et al., 2010; Mohr et al., 2013; Segura et al., 2014). However, the comparison of these approaches or the improvement of the compensation methodology used is beyond the scope of this study. Also the recently developed dual spot Aethalometer (AE33, Drinovec et al., 2015) allows for an improved and time-dependent loading compensation.

The compensated $b_{abs}$ is then converted into a BC mass using the mass absorption cross section (MAC)

$$BC = \frac{b_{abs}(\lambda)}{MAC(\lambda)} \qquad (5)$$

Usually nominal MAC values are used, to directly infer BC mass from the non-compensated $b_{ATN}$. These MAC values can be calculated from the parameters furnished by the Aethalometer manufacturer (Hansen, 2003) or are provided in the literature (e.g. Bond et al., 2013 and references therein) and include a $C$ value. Here MAC values are obtained empirically by comparing $b_{abs}$ with simultaneous measurements of EC from thermo-optical methods (e.g. Moosmüller et al., 2001, Bond et

al., 2013 and references therein), and the BC concentration is assumed to be identical to the EC concentration. From Equations 3 and 5 it is evident that empirically derived MAC values for absorption photometers strongly depend on the assumed $C$ value. Different $C$ values were previously empirically derived from instrumental comparisons and used to determine the absorption coefficient from Aethalometer measurements (e.g. Collaud Coen et al., 2010; Segura et al., 2014; Crilley et al., 2015). The separation of the $C$ value and the MAC is therefore relative to the methods used, and empirically

determined MAC values using Aethalometers should always be reported together with the applied $C$ values ($C = 2.14$ in our case).

### 2.2.2 Source apportionment using Aethalometer data

The spectral dependence of the absorption is described by the power law $b_{abs}(\lambda) \sim \lambda^{-\alpha}$ (Moosmüller et al., 2011), where $\alpha$ is the absorption Ångström exponent and consequently for a wavelength pair the

25 following relation can be derived:

$$\frac{b_{abs}(\lambda_1)}{b_{abs}(\lambda_2)} = \left(\frac{\lambda_1}{\lambda_2}\right)^{-\alpha} \qquad (6)$$

BC is a strong broadband absorber over the entire visible wavelength range (N-UV to N-IR) with only a weak spectral dependence ($\alpha$ for BC ~1). Traffic emissions mainly contain BC and basically no other light-absorbing compounds and consequently $\alpha$ for traffic emissions ($\alpha_{TR}$) ~1. Biomass-burning aerosols, on the other hand, contain additionally to BC a

30 substantial fraction of light-absorbing organic substances which strongly enhance the light absorption in the N-UV and blue part of the spectrum and have no contribution in the N-IR wavelength range resulting in an $\alpha$ for biomass-burning emissions ($\alpha_{wb}$) that is larger than $\alpha_{TR}$. Based on this, Sandradewi et al (2008a) developed a two-component model to apportion $b_{abs}$ measured with the Aethalometer at different wavelengths into a wood burning (WB) and a traffic (TR) contribution assuming that the total $b_{abs}$ is only influenced by these two sources.

$$b_{abs,total}(\lambda) = b_{abs,TR}(\lambda) + b_{abs,WB}(\lambda) \qquad (7)$$

This assumption is valid for Switzerland and other parts in Europe, where emissions from other sources (e.g. coal burning) are negligible. Using Equations 6-7 and the measured $b_{abs}$ at two different wavelengths a traffic and wood-burning contribution can be apportioned using the following equations:




$$\frac{b_{abs,TR}(\lambda_1)}{b_{abs,TR}(\lambda_2)} = \left(\frac{\lambda_1}{\lambda_2}\right)^{-\alpha_{TR}} \tag{8}$$

$$\frac{b_{abs,WB}(\lambda_1)}{b_{abs,WB}(\lambda_2)} = \left(\frac{\lambda_1}{\lambda_2}\right)^{-\alpha_{WB}} \tag{9}$$

$$b_{abs,WB}(\lambda_2) = \frac{b_{abs}(\lambda_1) - b_{abs}(\lambda_2)\cdot\left(\frac{\lambda_1}{\lambda_2}\right)^{-\alpha_{TR}}}{\left(\frac{\lambda_1}{\lambda_2}\right)^{-\alpha_{WB}} - \left(\frac{\lambda_1}{\lambda_2}\right)^{-\alpha_{TR}}} \tag{10}$$

$$b_{abs,TR}(\lambda_2) = \frac{b_{abs}(\lambda_1) - b_{abs}(\lambda_2)\cdot\left(\frac{\lambda_1}{\lambda_2}\right)^{-\alpha_{WB}}}{\left(\frac{\lambda_1}{\lambda_2}\right)^{-\alpha_{TR}} - \left(\frac{\lambda_1}{\lambda_2}\right)^{-\alpha_{WB}}} \tag{11}$$

The contributions of wood-burning and traffic to total BC ($BC_{WB}$ and $BC_{TR}$) are then derived via the corresponding MAC values ($MAC_{WB}$ and $MAC_{TR}$, respectively):

$$BC_{tot} = BC_{WB} + BC_{TR} = \frac{b_{abs,TR}(\lambda_2)}{MAC_{TR}(\lambda_2)} + \frac{b_{abs,WB}(\lambda_2)}{MAC_{WB}(\lambda_2)} \tag{12}$$

Consequently the ratio $BC_{TR}$ to total BC ($BC_{TOT}$) can be derived from the measured ratio $b_{abs}(\lambda_1)$ to $b_{abs}(\lambda_2)$ and assuming the ratio $MAC_{TR}(\lambda_2)$ to $MAC_{WB}(\lambda_2)$:

$$\frac{BC_{TR}}{BC_{tot}} = \frac{1}{1 - \frac{MAC_{TR}(\lambda_2)}{MAC_{WB}(\lambda_2)}\frac{1 - \frac{b_{abs}(\lambda_2)}{b_{abs}(\lambda_1)}\left(\frac{\lambda_1}{\lambda_2}\right)^{-\alpha_{TR}}}{1 - \frac{b_{abs}(\lambda_2)}{b_{abs}(\lambda_1)}\left(\frac{\lambda_1}{\lambda_2}\right)^{-\alpha_{WB}}}} \tag{13}$$

Using Equations 3-6, Equation 13 can be written as:

$$\frac{BC_{TR}}{BC_{tot}} = \frac{1}{1 - \frac{MAC_{TR}(\lambda_2)}{MAC_{WB}(\lambda_2)}\frac{1 - \frac{b_{ATN}(\lambda_2)\cdot C_{\lambda_1}\cdot R(\lambda_1, f_{\lambda_1}, ATN_{\lambda_1})}{b_{ATN}(\lambda_1)\cdot C_{\lambda_2}\cdot R(\lambda_2, f_{\lambda_2}, ATN_{\lambda_2})}\left(\frac{\lambda_1}{\lambda_2}\right)^{-\alpha_{TR}}}{1 - \frac{b_{ATN}(\lambda_2)\cdot C_{\lambda_1}\cdot R(\lambda_1, f_{\lambda_1}, ATN_{\lambda_1})}{b_{ATN}(\lambda_1)\cdot C_{\lambda_2}\cdot R(\lambda_2, f_{\lambda_2}, ATN_{\lambda_2})}\left(\frac{\lambda_1}{\lambda_2}\right)^{-\alpha_{WB}}}} \tag{14}$$

$\lambda_2$ has to be a wavelength in the N-IR range, where BC is the only light absorber, whereas $\lambda_1$ should be taken from the N-UV range where also organics contribute to the light absorption. In this model $\alpha_{WB}$ and $\alpha_{TR}$ have to be assumed a priori or

determined comparing the contributions of $BC_{TR}$ and $BC_{WB}$ to other techniques which apportion BC or EC into those two sources (e.g. $^{14}$C measurements). Additional uncertainties may arise from the compensation factors applied to the attenuation coefficients. In this study, a fixed $C_\lambda$ value was used for the multi-scattering correction (Sect. 2.2.1) and thus the ratio $C_{\lambda,1}/C_{\lambda,2}$ becomes unity in Eq. 14. This is justified and introduces very little uncertainty, as the wavelength dependence of the $f$ and $C$ values across the range $\lambda$ = 470-950 nm was reported to be less than 10% and 12%, respectively, for the

Aethalometer model AE31 (Weingartner et al., 2003; Sandradewi et al., 2008c; Segura et al., 2014). If data from other photometer models, which exhibit a wavelength dependence of the C value, are used for the source apportionment, the correct ratio $C_{\lambda,1}/C_{\lambda,2}$ must be used in Eq. 14 to ensure consistency of the Aethalometer model parameters. The loading compensation factor $R(f_\lambda, ATN_\lambda)$ depends on wavelength, even if $f(\lambda)$ is independent of wavelength, since the ATN depends considerably on the wavelength. Nevertheless, uncertainties in the $BC_{TR}$ to BC ratio associated with the filter loading

compensation can be kept small by carefully determining the $f$ values, following the approach in Weingartner et al. (2003) or Sandradewi et al. (2008c). The Aethalometer AE33 measures the compensation parameters and therefore the compensation is performed on-line. The precision of this compensation can be checked using the BC(ATN) or $b_{abs}$(ATN) analysis (Drinovec et al., 2015).

Sandradewi et al. (2008a) and subsequent studies that used the Aethalometer model utilized the same MAC for traffic

($MAC_{TR}$) and wood burning ($MAC_{WB}$) emissions at the N-IR wavelength, based on the fact that MAC values for freshly generated BC, were previously found to fall within a relatively narrow range (Bond and Bergstrom, 2006 and references therein). However, MAC values depend on particle size, morphology and mixing state and thus different values for biomass



burning and traffic emissions may be possible. Therefore, we assess the ratio of $MAC_{TR}$ to $MAC_{WB}$ for our dataset in Sect. 3.1 and 3.2.1.

Sandradewi et al. (2008a) and many other studies used 470 nm and 950 nm as N-UV and N-IR wavelengths, respectively. However, also other combinations of wavelengths have been used (e.g. 370 and 880 nm, or 470 nm and 880 nm, see Perron
et al. (2010), Herich et al. (2011) and Fuller et al. (2014)), especially in studies that performed Aethalometer measurements with the two-wavelength instrument (370 nm and 880 nm, model AE22, Magee Scientific). Therefore, we will also investigate the sensitivity of the Aethalometer model using different wavelength combinations.

### 2.3 Radiocarbon analysis

#### 2.3.1 Separation of the carbonaceous particle fractions

Two different methods to isolate EC for the [14]C analysis were used. For the samples from the [14]C project Switzerland, the Swiss_4S protocol was applied for the EC isolation using a Sunset OC/EC analyser as described by Zhang et al. (2012). This approach is optimized such that biases in the [14]C result of EC due to OC charring or losses of the least refractory EC during the OC removal are minimized. In brief, to minimize positive artefacts from OC charring, water-soluble OC (WSOC) is first eliminated by a water extraction and the remaining water-insoluble OC (WINSOC) is then removed using the Sunset
analyser by a thermal treatment in three steps: (1) 375 °C for 150 s in pure Oxygen ($O_2$); (2) 475 °C for 180 s in $O_2$; (3) 450 °C for 180 s followed by 180 s at 650 °C in helium. Finally, in step four EC is isolated by the combustion of the remaining carbonaceous material at 760 °C for 150 s in $O_2$. The evolving $CO_2$ is separated from interfering gaseous products, cryo-trapped and sealed in glass ampoules for [14]C measurements. By using the Sunset analyser, which monitors the transmission of light through the filter with a laser during the combustion, the quantification of OC charring and EC losses
before step 4 is achieved. For the samples of the [14]C project Switzerland, charred OC only contributed ~5% to EC recovered in step four and on average 74 ± 11% of the EC was recovered for the [14]C measurement. Charring OC of a given thermal step is quantified as the difference of the maximum ATN and the initial ATN normalized to the initial ATN. The EC recovery is related to the loss of EC during the first three steps and is defined as the ratio between the initial ATN of the laser signal through the filter before step one of the thermal treatment and the ATN before step four (Zhang et al., 2012).

EC from samples collected during the campaigns in ROV, MOL, MAS, REI, ZUR in 2005 and 2006 was isolated for the [14]C analysis using the THEODORE system and the approach described by Szidat et al. (2004). In brief, after removal of WSOC by water extraction, WINSOC was evaporated during 4 h in a muffle furnace in air at 375 °C. EC was finally combusted in the THEODORE system at 640 °C for 10 min with $O_2$. The evolving $CO_2$ was recovered in the same manner as described above. The EC recovery for these samples was estimated by Zhang et al. (2012) and was on average 60 ± 12%. The [14]C
results of EC were corrected to 100% EC recovery (see section 2.4.3 below) obtained with the THEODORE and the Swiss_4S method were previously found to agree within the uncertainties (see Zhang et al., 2012).

#### 2.3.2 Radiocarbon measurement

The analysis of the [14]C content in the $CO_2$ from the separated EC fraction collected as described above was carried out with the **MI**ni radio**CA**rbon **DA**ting **S**ystem, MICADAS (Synal et al., 2007) at the Swiss Federal Institute of Technology (ETH)
Zürich and the Laboratory for the Analysis of Radiocarbon with AMS (LARA), University of Bern (Szidat et al., 2014) using a gas ion source (Ruff et al., 2010; Wacker et al., 2013). The results of the [14]C measurement are presented as fraction of modern ($f_M$) denoting the [14]C/[12]C content of the sample related that of the reference year 1950 (Stuiver and Polach, 1977). The $f_M$ values are corrected for $\delta^{13}C$ fractionation and for [14]C decay between 1950 and the year of measurement (Wacker et al., 2010). The $f_M$ measurement uncertainty for the EC samples from the [14]C project Switzerland and ROV, MOL, MAS,
REI, ZUR from 2005 and 2006 is ~2% (Zotter et al., 2014) and ~3% respectively (Zhang et al., 2012).





### 2.3.3 Determination of the non-fossil fraction of EC

As shown above (see Sect. 2.2), on average only 74 ± 11% and 60 ± 12% of the total EC (EC yield) was isolated for the [14]C measurement of the samples from the [14]C project Switzerland and ROV, MOL, MAS, REI, ZUR from 2005 and 2006, respectively. However, Zhang et al. (2012) showed that $f_M$ values are lower for lower EC yields suggesting that the EC that

is removed before step four (the step in which EC is recovered for the [14]C measurement), is mainly from biomass burning due to its lower thermal stability (Zhang et al., 2012). Therefore, an extrapolation of the measured EC $f_M$ values to 100% EC yield was applied to account for this underestimation of $f_M$ (Zhang et al., 2012). This method was applied to all samples discussed here, and the detailed description of the procedure used for the samples from the [14]C project Switzerland and ROV, MOL, MAS, REI, ZUR from 2005 and 2006 can be found in Zotter et al. (2014) and Zhang et al. (2012), respectively.

The $f_M$ of contemporary carbon including biogenic sources and biomass burning ($f_{M,bio}$ and $f_{M,bb}$, respectively) is characterised by values of 1 whereas $f_M$ is equal to 0 for fossil sources due to the decay of [14]C with a half-life of 5730 years. Due to the nuclear weapon tests in the 1950s and 1960s, however, the atmospheric [14]C content increased and $f_M$ exhibits values >1 (Levin et al., 2010). Therefore, $f_M$ values for EC were converted into non-fossil fractions ($f_{NF,EC}$) (Szidat et al., 2006) using a reference value. Since biomass burning is the only non-fossil source of EC (neglecting possible small

contributions from bio-fuels) this reference value is equal to $f_{M,bb}$ and was estimated using a tree-growth model as described in Mohn et al. (2008) including 10-year, 20-year, 40-year, 70-year and 85-year old trees with weight fractions of 0.2, 0.2, 0.4, 0.1, and 0.1, respectively, harvested three years before aerosol sampling. Values of 1.140, 1.135, 1.127, 1.123, 1.119, 1.114 and 1.106 were calculated and consequently used to correct the $f_M$ values extrapolated to 100% EC yield from samples collected in 2005, 2006, 2008, 2009, 2010, 2011 and 2012, respectively. The final uncertainties for $f_{NF,EC}$ (~5 % and ~6% for

samples from the [14]C project Switzerland and ROV, MOL, MAS, REI, ZUR from 2005 and 2006, respectively) are derived from an error propagation and include all the individual uncertainties of $f_M$ (measurement uncertainty, extrapolation to 100% EC yield) and $f_{M,bb}$ (Zotter et al., 2014).

### 2.4 Elemental carbon measurement

The EC concentrations on samples from the [14]C project Switzerland (see Table 1) were measured using a thermo-optical
OC/EC analyser (Model 4L, Sunset Laboratory Inc., USA), equipped with a non-dispersive infrared (NDIR) detector following the thermal-optical transmittance protocol (TOT) EUSAAR2 (Cavalli et al., 2010). EC concentrations from the campaigns in ROV, MOL, MAS, REI, ZUR in 2005 and 2006 (see Table 1) are not included for the MAC calculations, since in earlier campaigns they were not measured or obtained with a different TOT protocol. We assigned a high uncertainty of 25% for all measured EC concentrations to account for possible differences between different TOT protocols (Schmid et al.,

2001). It should be noted that only the MAC determination is affected by the uncertainty of the EC concentrations whereas the evaluation of the choice of $\alpha_{WB}$ and $\alpha_{TR}$ using the fossil fraction of EC is influenced by the combined uncertainty of the [14]C measurement of EC, the extrapolation of $f_{M,EC}$ to 100% EC yield and the bomb peak correction which was on average only 5–6% (see Sect. 2.3). No EC was detected on blank filters and consequently no blank correction was necessary (see also Zotter et al., 2014).

### 35 2.5 Additional data

Nitrogen oxides (NO$_x$) are routinely measured at the NABEL stations ZUR, MAG and PAY using reference instrumentation with molybdenum converters according to valid European standards (EMPA, 2013). Since no large sources of NO$_x$ (e.g. fossil fuel power plants) are present in Switzerland besides traffic, NO$_x$ will be used here for the comparison with BC$_{TR}$ (see Sect. 3.3 below).

Levoglucosan, a thermal degradation product of cellulose and thus a tracer for primary emissions of organic aerosol from biomass burning and often used to estimate OC mass from this source (Gelencsér et al., 2007), was also measured on 52





samples presented in this study. A description of the measurement details can be found in the corresponding references as listed in Table 1. Levoglucosan data is available for most of the samples from winter 2005 and 2006 from ROV, MOL, REI, MAS and ZUR (n = 27) as well as from the winter 2008/2009 for ZUR, MAG and PAY (n = 8) from the [14]C project Switzerland (see Zotter et al., 2014). In addition, data from these three stations (n = 17) with parallel Aethalometer

measurements available were also taken from Herich et al. (2011). Levoglucosan data will be used here for the comparison with $BC_{WB}$ (see Sect. 3.3 below). As photochemical degradation of levoglucosan was previously observed under summertime conditions (Kessler et al., 2010; Hennigan et al., 2011), spring and summer levoglucosan data from ZUR are not used here.

### 3 Results and discussion

**3.1 MAC determination**

MAC values are determined empirically by comparing $b_{abs}$ with EC thermo-optical measurements (see Fig. 2a). $b_{abs}$ at 880 nm and EC are strongly correlated ($r = 0.86$) and the geometric mean of the MAC at 880 nm was found to be 11.8 $m^2g^{-1}$ (9.2-15.1 $m^2g^{-1}$), similar to values obtained in Herich et al. (2011) for ZUR (10.0 $m^2g^{-1}$), PAY (13.2 $m^2g^{-1}$) and MAG (9.9 $m^2g^{-1}$) for a 2-year dataset (note that the MAC values reported in this study as well as that by Herich et al. (2011) both

apply for EC mass based on the thermal-optical transmittance protocol EUSAAR 2 and absorption coefficients inferred from Aethalometer AE31 data with assuming a $C$ value of 2.14). No systematic year-to-year or station-to-station variations in the MAC values at 880 nm are observed. While the MAC values determined at SIS are lower on average, they remain within the previously reported range, and given the relatively modest number of samples, this observation cannot be generalized. It should be noted that MAC values depend on the aerosol mixing state, size and morphology (see e.g. Bond and Bergstrom,

2006), and empirically derived MAC values also depend on the limitations of the measurement techniques used to determine $b_{abs}$. Results of our study would translate to ~9.7-10.0 $m^2 g^{-1}$ at 637 nm when recalculating our MAC values from a wavelength of 880 nm to 637 nm with an AAE of 0.9-1.0 and if a $C$ value of 3.5 instead of 2.14 was assumed. This is in good agreement with the average MAC value of 10.0 $m^2 g^{-1}$ at 637 nm reported by Zanatta et al. (2016) for 9 European background sites, who also used the EUSAAR-2 protocol for EC mass and either Multi Angle Absorption Photometers,

Particle Soot Absorption Photometers or Aethalometers with assuming $C = 3.5$ for the absorption coefficient. Deviations from other previously reported MAC values at similar wavelengths (5-26 $m^2 g^{-1}$, Liousse et al., 1993; Bond and Bergstrom, 2006, Genberg et al., 2013) can be due to different methods used to determine EC and absorption coefficient and/or possible differences in BC size and mixing state.

Only few studies attempted the empirical determination of MAC values for biomass burning and traffic BC emissions using

ambient measurements (e.g. Laborde et al., 2013, Bond et al., 2013 and references therein). Since the ratio of $MAC_{TR}$ to $MAC_{WB}$ at the N-IR wavelength is needed in the Aethalometer model (see Equation 13), it is important to assess possible differences between $MAC_{TR}$ and $MAC_{WB}$. Sandradewi et al. (2008a) and all other studies that applied the Aethalometer model assumed, implicitly or explicitly, a $MAC_{TR}$ to $MAC_{WB}$ ratio of unity at 880 nm. Having an independent measurement for the relative contributions of traffic and wood-burning to total EC from the [14]C measurements, allows to test this

assumption by plotting the MAC values at 880 nm against the corresponding relative traffic contribution to EC ($EC_F/EC$) obtained with the [14]C measurements (see Fig. 2b). No correlation between the two parameters was found, indicating that it is justified to simplify the Aethalometer model (Eq. 12) and set the ratio of $MAC_{TR}$ to $MAC_{WB}$ at the N-IR wavelength to unity. This is in agreement with Herich et al. (2011) who did not find differences in MAC for the stations ZUR, MAG and PAY for a 2-year dataset between summer and winter, where there is a large seasonality in the relative wood-burning

contribution. The variability in Fig. 2b is due to day-to-day and station-to-station variability but could to some degree also originate from different size cuts ($PM_{10}$ or $PM_1$ and $PM_{2.5}$) of the filter samplers and Aethalometer measurements for some



campaigns (see Table 1). Alternatively, the ratio of $MAC_{TR}$ to $MAC_{WB}$ at the N-IR wavelength can be used as a third free parameter, besides $\alpha_{TR}$ and $\alpha_{WB}$, when fitting the Aethalometer model (Eq. 12) against a dataset of independent $EC_F/EC$ measurements. We tested this for the data set of this study and obtained a $MAC_{TR}$ to $MAC_{WB}$ ratio of 0.97, which confirms the finding of Fig. 2b. Therefore, in the following we will use a fixed $MAC_{TR}$ to $MAC_{WB}$ ratio of 1 in the Aethalometer

model.

### 3.2 Application and evaluation of the Aethalometer model

#### 3.2.1 Best $\alpha_{TR}$ and $\alpha_{WB}$ pair, and analyses of uncertainties and biases

Independent measurements of the contribution of wood burning and traffic to BC (or EC) are often not available; therefore in most studies a single $\alpha_{TR}$ and $\alpha_{WB}$ pair is usually used in the Aethalometer model, derived from previous work. However,

$\alpha_{WB}$ and $\alpha_{TR}$ may be highly variable, depending on the combustion conditions and efficiency, fuel type and aerosol aging (Lack et al., 2013; Saleh et al., 2013; Zhong and Jang, 2014; Sharpless et al., 2014; Saleh et al., 2014; Kirchstetter et al., 2004; Bond and Bergstrom, 2006 and references therein; Herich et al., 2011; Garg et al., 2016). In this section we use $EC_F/EC$ values from [14]C measurements to determine the best combination of $\alpha_{TR}$ and $\alpha_{WB}$ and assess the performance of the Aethalometer model using this single pair of $\alpha$ values. In practice, the best pair of $\alpha$ values is determined by fitting

Equation 13 against $EC_F/EC$ from the [14]C analyses using the ratio $b_{abs,470}/b_{abs,950}$ from the Aethalometer as independent variable (and assuming $MAC_{TR,950}/MAC_{WB,950} = 1$, as justified in Sect. 3.1). We use a least square fitting weighted by the inverse number of data points in $EC_F/EC$ bins of 0.1 as most of the data presented in this study fall within a range of $EC_F/EC = 0.4–0.6$. The absorption Ångström exponents $\alpha_{TR}$ and $\alpha_{WB}$ that fit best our data were found to be 0.90 and 1.68, respectively. The same $\alpha$ values were obtained when $MAC_{TR}/MAC_{WB}$ was included as a third fitting parameter, because the

best fit MAC ratio is 0.97, which is virtually equal to unity (see also Sect. 3.1).

$BC_{TR}/BC$ at 950 nm, derived with above best fit Aethalometer model parameters, and $EC_F/EC$ are well correlated ($r = 0.81$ see Fig. 3a) and the fitting residuals ($\Delta BC_{TR}/BC = BC_{TR}/BC–EC_F/EC$, Fig. 3b) are normally distributed with only a minor positive bias of 1.6%. We estimate that the precision of the model ($\Delta BC_{TR}/BC$) is on average 9.3% in our case, using the standard deviation ($\sigma$) of the Gaussian fit of $\Delta BC_{TR}/BC$ in Fig. 3b. This indicates that the Aethalometer model reproduces

reasonably well the [14]C results for all stations investigated in this study using our best estimate of a single $\alpha_{WB}$ and $\alpha_{TR}$ pair. Since this analysis includes data from urban stations as well as from spring and summer this shows that the Aethalometer model also works for other areas than for polluted alpine valleys in winter. It should be noted that the determination of $BC_{TR}/BC$ using the fitted $\alpha$ values cannot be more accurate than the uncertainty of $EC_F/EC$. The estimated $\Delta BC_{TR}/BC$ is affected by (1) random measurement uncertainties of $EC_F/EC$ and $b_{abs,470}$ and $b_{abs,950}$ and (2) day-to-day and station-to-station

variability in $\alpha_{WB}$ and $\alpha_{TR}$ values. Investigating the effect of a $MAC_{TR,950}/MAC_{WB,950}$ different from one ($MAC_{TR}/MAC_{WB} = 0.7–1.3$) it is evident that there is no large influence on $\alpha_{WB}$ (1.66–1.71), $\alpha_{TR}$ (0.8–0.95), the mean bias (0.2–2.4%) and $\Delta BC_{TR}/BC$ (9.4–9.9%). This further justifies fixing the MAC ratio at unity when applying the Aethalometer model.

Without an alternative method for the source apportionment of EC or BC, the determination of $\alpha$ values and related

uncertainties are unattainable. Therefore, we determined the distribution of $\alpha_{WB}$ values for our dataset and investigated if there are other combinations of $\alpha_{TR}$ and $\alpha_{WB}$ that yield similarly acceptable agreement with [14]C data. For this purpose, Equation 13 was solved for $\alpha_{WB}$:

$$\alpha_{WB} = \frac{-1}{ln\left(\frac{\lambda_1}{\lambda_2}\right)} \cdot ln\left(\frac{b_{abs}(\lambda_1)}{b_{abs}(\lambda_2)} + \frac{\frac{MAC_{TR}(\lambda_2)}{MAC_{WB}(\lambda_2)}\left(\left(\frac{\lambda_1}{\lambda_2}\right)^{-\alpha_{TR}} - \frac{b_{abs}(\lambda_1)}{b_{abs}(\lambda_2)}\right)}{1 - \frac{EC}{EC_F}}\right) \tag{15}$$





This makes it possible to analytically calculate $\alpha_{WB}$ for every single data point, if a fixed $\alpha_{TR}$ is assumed and setting $MAC_{TR}$ to $MAC_{WB}$ to unity. $\alpha_{WB}$ values were calculated for three different $\alpha_{TR}$ values of 0.9, 1.0 and 1.1, which represent the range previously used in the literature. The resulting three $\alpha_{WB}$ distributions are displayed in Fig. 4. It is evident that an increase in $\alpha_{TR}$ by 0.1 results in a concurrent decrease in $\alpha_{WB}$ by 0.1. This covariance between $\alpha_{TR}$ and $\alpha_{WB}$ implies that using

combinations of $\alpha_{TR}$ and $\alpha_{WB}$ randomly altered (e.g. ± 0.1) from the best $\alpha$ pair could result in high $\Delta BC_{TR}/BC$. No clear station-to-station or season-to-season differences in $\alpha_{WB}$ were observed (see Table 2), though the number of samples from each station inspected here is limited for such analysis.

Investigating the different distributions in Fig. 4 only the range of $\alpha$ combinations resulting in the best agreement between Aethalometer model and $^{14}C$ results of all individual data points can be obtained but it is not possible to determine other

single $\alpha$ pairs representative for the entire data set. To investigate the bias in $BC_{TR}/BC$ with respect to $EC_F/EC$ ($\Delta BC_{TR}/BC$) due to deviations of $\alpha_{TR}$ and $\alpha_{WB}$ ($\Delta\alpha_{TR}$ and $\Delta\alpha_{WB}$, respectively) from the best $\alpha$ pair, Equation 13 was differentiated with respect to both parameters as a function of $BC_{TR}/BC$. From Fig. S1 (see supporting information) it is evident that $\Delta BC_{TR}/BC$ is dependent on $BC_{TR}/BC$: for high and low values of the latter, $\Delta BC_{TR}/BC$ is mainly driven by $\Delta\alpha_{TR}$ and $\Delta\alpha_{WB}$, respectively. A $\Delta\alpha_{WB}$ of 0.1 yields a max. $\Delta BC_{TR}/BC$ of 17% and a $\Delta\alpha_{WB}$ of 0.2 already results in a max. $\Delta BC_{TR}/BC$ of 33%. On the other

hand, a $\Delta\alpha_{TR}$ of 0.2 results in only a max. $\Delta BC_{TR}/BC$ of 19%. Exploring $\Delta BC_{TR}/BC$ for different $\alpha$ combinations ($\alpha_{TR} = 0.9$-1.1 and $\alpha_{WB} = 1.4$-2.2) as a function of $EC_F/EC$ (see Fig. 5) it is evident that other $\alpha$ pairs exist yielding low $\Delta BC_{TR}/BC$ but, in contrast to the best $\alpha$ pair ($\alpha_{TR} = 0.9$ and $\alpha_{WB} = 1.68$ as obtained in Sect. 3.2.1) not over the entire range of $EC_F/EC$ found in this study. Especially for $EC_F/EC < 30\%$ almost all $\alpha$ combinations, except the best pair, lead to a significant over- or under-estimation of $BC_{TR}/BC$ compared to $EC_F/EC$. Considering the $1\sigma$ confidence interval of $\Delta BC_{TR}/BC$ (minimum of -

0.6% and maximum of 14%) as acceptable deviation from $EC_F/EC$ also combinations of $\alpha_{TR} = 0.8$ (see Fig. S2) and 1.0 and $\alpha_{WB} = 1.8$ and 1.6, respectively, are possible but only for a range of $EC_F/EC$ between ~40% and ~85%. The $\alpha$ pair obtained by Sandradewi et al. (2008a) ($\alpha_{TR} = 1.1$ and $\alpha_{WB} = 1.86$) who first used the Aethalometer model results in a constant positive bias of $BC_{TR}/BC$ compared to $EC_F/EC$ and does not even fall within the $3\sigma$ confidence interval of $\Delta BC_{TR}/BC$ (upper range ~30%). Furthermore, for $\alpha_{TR}$ of 1.1 only a very narrow range of $EC_F/EC$ (spanning maximum 20%) exists resulting in

$\Delta BC_{TR}/BC$ within the $1\sigma$ confidence interval. In addition, from Fig. 5 it is also evident that almost no $\alpha_{WB}$ previously used in literature (1.8-2.2) would yield $\Delta BC_{TR}/BC$ within the $1\sigma$ confidence for our data set indicating that lower values of $\alpha_{WB}$ should be used in the future in the Aethalometer model.

### 3.2.2 Evaluation of the Aethalometer model against external data

A further evaluation of the source apportionment results of the Aethalometer model was carried out by comparing $BC_{WB}$ and

$BC_{TR}$ calculated with the best $\alpha_{TR}$ and $\alpha_{WB}$ pair (0.90 and 1.68, respectively) with other markers for traffic and biomass burning emissions. Fig. 6a presents the correlation of $NO_x$, considered to be from traffic emissions, with $BC_{TR}$, both averaged to 24 h from the NABEL stations PAY, MAG and ZUR for the winter seasons 2009–2012, where Aethalometer and $NO_x$ measurements were performed continuously for several years (see Sect. 2.5 and Table 1). Good correlations are found ($r = 0.76$–0.83) and all stations exhibit similar slopes (24.7-30.7 ppb/μg C m$^{-3}$) and small axis intercepts (Fig. 6a).

These slopes are comparable to London (18-28 ppb/μg C m$^{-3}$, Liu et al., 2014), Grenoble (33 ppb/μg C m$^{-3}$, Favez et al., 2010) and several other locations in Switzerland (32 ppb/μg C m$^{-3}$, Zotter et al., 2014). Levoglucosan obtained on filter samples collected during the winter season and $BC_{WB}$ were also found to be well correlated ($r = 0.77$, see Fig. 6b) with also only a minor intercept. The slope obtained here (1.08) is also similar to other locations (1.0 for several other locations in Switzerland (Zotter et al., 2014), 0.76 for three sites in Austria (Caseiro et al., 2009), 1.12 in the Po-Valley (Gilardoni et al.,

2009, Piazzalunga et al., 2011) and 1.68 in Grenoble (Favez et al., 2010)).





### 3.2.3 Comparison of $\alpha_{TR}$ and $\alpha_{WB}$ with literature values

The $\alpha_{TR}$ value obtained here (0.9) is lower than the values used in Sandradewi et al. (2008a) and many other studies (1–1.1, Favez et al., 2010; Crippa et al., 2013; Mohr et al., 2013). However, our findings are in agreement with those reported in Herich et al. (2011) showing that ambient $\alpha$ values in ZUR, MAG and PAY in summer, when hardly any biomass burning influence is expected, are around ~0.9. Herich et al. (2011) consequently used then this value as $\alpha_{TR}$ in the Aethalometer model. Also Fuller et al. (2014) determined a value below 1 ($\alpha_{TR}$ of 0.96) for London.

The $\alpha_{WB}$ values obtained in this study are consistent with those reported from smog chamber experiments for fresh and aged biomass-burning emissions (1.63 ± 0.32, Saleh et al., 2013), but are significantly lower than the values from Sandradewi et al. (2008a) often used by other source apportionment studies, i.e. 1.8–2.2 (Sandradewi et al., 2008a; Sandradewi et al., 2008b; Favez et al., 2010; Perron et al., 2010; Herich et al., 2011; Harrison et al., 2012; Crippa et al., 2013; Harrison et al., 2013; Mohr et al., 2013). Note that Sandradewi et al. (2008a) determined their best pair of $\alpha$ values ($\alpha_{TR} = 1.1$ and $\alpha_{WB} = 1.86$ calculated with $b_{abs,470}$, $b_{abs,950}$ and $MAC_{TR}/MAC_{WB} = 1$) by optimizing the ratio of the total fossil carbonaceous matter ($CM_F/CM$ instead of $EC_{TR}/EC$) obtained from the $^{14}C$ measurements (see Sect. 3.4). Furthermore, Sandradewi et al. (2008a) did not account for the slight underestimation of biomass burning EC as discussed in Zhang et al. (2012). Applying the approach presented in Sect. 3.2.2 to the data in Sandradewi et al. (2008a), using their value for $\alpha_{TR}$ and $MAC_{TR}/MAC_{WB}$, yields $\alpha_{WB}$ of 1.64 and 1.72 with and without extrapolation of $EC_F/EC$ to 100% EC yield, respectively, which is very similar to the values obtained in this study. Meanwhile, applying the pair of $\alpha$ values previously used from Sandradewi et al. (2008a) to determine $BC_{TR}/BC$ from our data results in a mean positive bias ($\Delta BC_{TR}/BC = 18\%$), and therefore the use of this pair is not recommended in future studies.

Recently, Garg et al. (2016) investigated ambient $\alpha$ values in India for various biomass combustion plumes including paddy- and wheat-residue burning, leaf litter, and garbage burning as well as traffic plumes. They found $\alpha$ values down to one for flaming biomass combustion and $\alpha > 1$ for older vehicles operating with poorly optimized engines and that $\alpha$ was mostly determined by the combustion efficiency. Therefore, if more than two tightly regulated BC sources, with well constrained combustion efficiencies are present, the $\alpha$ values might be different and additional evaluations of the choice of $\alpha_{WB}$ and $\alpha_{TR}$ in the Aethalometer model using a reference method are needed as well.

### 3.2.4 Sensitivity due to different wavelength combinations

As different pairs of N-UV and N-IR wavelengths (470 & 950 nm, 470 & 880 nm and 370 & 880 nm, see Perron et al. (2010), Herich et al. (2011) and Fuller et al. (2014)) have been previously used in literature, we investigated the sensitivity of the Aethalometer model using different wavelength combinations by performing the same analysis as presented in Sect. 3.2.1 with different N-UV and N-IR pairs. In contrast to 470 & 950 nm, no physically meaningful values for $\alpha_{TR}$ could be obtained for the other combinations by fitting Equation 13 against $EC_F/EC$ (see Table 4). Consequently, $\alpha_{TR}$ was set to 0.9 to infer $\alpha_{WB}$ for the combinations 470 & 880 nm, 370 & 950 nm and 370 & 880 nm. As shown in Table 4 different $\alpha_{WB}$ values for these wavelength pairs were obtained than for 470 & 950 nm. Especially using 370 nm as the N-UV wavelength yielded a significantly higher $\alpha_{WB}$ (2.09) than using 470 nm (1.68 and 1.75 for 470 & 950 nm and 470 & 880 nm, respectively). It has been reported that $\alpha$ is wavelength dependent (e.g. Bond and Bergstrom, 2006), and might be more affected by fuel type, combustion and atmospheric processes in the N-UV than in the visible part of the spectrum (Sandradewi et al., 2008c). Consequently, $\alpha_{WB}$ can be different for different wavelength pairs. However, for all combinations, especially with 370 nm as N-UV wavelength, the mean residuals of $BC_{TR}/BC$ compared to $EC_F/EC$, were higher than using the 470 & 950 nm combination (see Table 4a). Next, $BC_{TR}/BC$ was calculated with the best pair of $\alpha$ values ($\alpha_{WB} = 1.68$ and $\alpha_{TR} = 0.90$ as obtained in Sect. 3.2.1) for the different wavelength combinations. It is evident that using 370 nm as N-UV wavelength $BC_{TR}/BC$ exhibits an inferior agreement with $EC_F/EC$ (see Table 4b). $\Delta BC_{TR}/BC$ exhibits larger values, there is a significant number of negative points for $BC_{TR}/BC$ and the correlations with $EC_F/EC$ are weaker. On





the other hand, similar $\Delta BC_{TR}/BC$ and hardly any negative $BC_{TR}/BC$ values are found for the wavelength combination 470 & 880 nm.

Uncertainties in the source apportionment results using the Aethalometer model due the use of different wavelength pairs are usually not considered and often the same $\alpha_{WB}$ and $\alpha_{TR}$ values are used with different wavelength combinations. However, as

shown here the choice of the wavelengths, especially the one in the N-UV range and $\alpha_{WB}$ are not independent. Since 1) it was previously shown that adsorption of VOCs on the filter tape of the Aethalometer can occur which possibly influences the absorption measurement with the 370 nm channel (Vecchi et al., 2014), 2) light absorbing SOA, other absorbing non-BC combustion particles and atmospheric processing affects lower wavelengths more than higher ones and 3) our results indicate an inferior agreement of $BC_{TR}/BC$ with $EC_F/EC$ using 370 nm as N-UV wavelength we therefore recommend using 470 nm

as the N-UV wavelength in the Aethalometer model while the choice between 950 nm and 880 nm in the N-IR is less critical.

### 3.2.5 High time resolution data

Since Aethalometers measure with high time resolutions (e.g., model AE31 down to 2 min and the new model AE33 down to 1 sec) the investigation of the temporal behavior of $BC_{TR}$ and $BC_{WB}$ is possible (see e.g., Herich et al. 2011). Fig 7 and

Fig S3 show the diurnal cycles for the stations MAG, PAY and ZUR including continuous data from the entire years 2009 to 2012. It is evident that the Aethalometer can also be applied to high time resolution data and the expected temporal behavior of the sources can be resolved. The contribution of $BC_{WB}$ is high in winter and during the night, with only small differences between weekends and weekdays. Furthermore, $BC_{TR}$ exhibits a clear traffic peak in the morning during week days whereas during weekends this increase is not evident or only small.

We note that $BC_{WB}$ also follows $BC_{TR}$, with an evident increase during morning hours. This increase is statistically larger than our uncertainties (14-18% in winter and 30-75% in summer). This indicates that there is some false attribution of $BC_{TR}$ and $BC_{WB}$ most probably due to the constant a priory assumed pair of $\alpha_{WB}$ and $\alpha_{TR}$. By applying different $\alpha$ combinations for ZUR ($\alpha_{WB} = 1.68$ and $\alpha_{TR} = 0.90$, $\alpha_{WB} = 1.68$ and $\alpha_{TR} = 1.1$, $\alpha_{WB} = 1.9$ and $\alpha_{TR} = 0.90$ as well as $\alpha_{WB} = 1.9$ and $\alpha_{TR} = 1.1$) this false attribution between $BC_{TR}$ and $BC_{WB}$ during the morning peak disappears (see Fig. S4), indicating that a higher $\alpha_{TR}$

would be more representative of fresh traffic emissions in the case of ZUR. Since the evaluation of $\alpha$ combinations presented in this paper is based on longer sampling times and mostly winter data (16h to 40h, see Table 1), caution should be taken when applying the Aethalometer model with the best $\alpha$ pair found here to high time resolution data, especially for low BC concentrations and rush hours. Similar studies with higher time resolutions, and for BC concentrations, like in summer, are necessary for a further evaluation of the Aethalometer model.

### 3.3 Traffic and wood burning contributions to EC and BC

The relative traffic contribution as apportioned by the Aethalometer model ($BC_{TR}/BC$) and the [14]C analysis ($EC_F/EC$) of BC and EC, respectively, is often >50% (see Fig. 3a). However, since hardly any $EC_F/EC$ values, except results from the summer season, are above 70% and the average of $EC_F/EC$ over all winter samples is $52 \pm 17\%$, it is evident that also wood burning emissions account for a large fraction of EC (and thus BC) during winter in Switzerland. The traffic contributions

for winter samples range from 7–82% and 14–84% for of BC and EC, respectively. The lowest values ($31 \pm 23\%$ and $36 \pm 17\%$ for $BC_{TR}/BC$ and $EC_F/EC$, respectively) were found at ROV which is most likely due to a combination of topography (ROV is located in an Alpine valley), local meteorology (often persistent inversions with low mixing heights) and emissions (high local wood burning influence) (Alfarra et al., 2007; Szidat et al., 2007; Lanz et al., 2008; Sandradewi et al., 2008a; 2008c; Herich et al., 2014; Zotter et al., 2014). The samples from ZUR, the largest city of Switzerland, collected

during spring and summer clearly show the highest fossil contributions with an average of $81 \pm 10\%$ and $80 \pm 7\%$ and the highest value of 92% and 85% for $BC_{TR}/BC$ and $EC_F/EC$, respectively.



Investigating the diurnal cycles of $BC_{WB}$ it is evident that the concentrations are high in winter, especially in MAG and during night time, with no or only small differences between weekends and weekdays. Concentrations in summer are lower but non-negligible, with significantly (paired t-test, significance level of 0.05) higher concentrations in Zurich compared to the other locations, especially during night times on weekends. This suggests an additional source of brown carbon in ZUR,

likely related to primary emissions enhanced with urban activities, during weekends, and with higher emissions in an increasingly shallower night-time boundary layer. Contribution of secondary processes to the brown carbon background concentrations observed at all sites cannot be excluded. $BC_{TR}$ concentrations for week days are significantly higher in winter compared to summer and also for weekends in MAG (t-test, significance level of 0.05). In contrast, in ZUR and PAY, average $BC_{TR}$ weekend concentrations are very similar. Weekday $BC_{TR}$ concentrations exhibit a clear morning peak for all

stations and seasons, which is less evident on weekends. The lowest concentrations are found in PAY and the highest in MAG in winter. In summer, $BC_{TR}$ is highest in ZUR days).

**3.4 Traffic and wood burning contributions to PM**

It has been attempted to also apportioned the total carbonaceous material (CM) to wood-burning ($CM_{WB}$) and traffic ($CM_{TR}$) (e.g., Sandradewi et al., 2008a) according to the following equations:

$CM = OM + BC$                                                         (16)

$CM = CM_{TR} + CM_{WB} + CM_{other} = c_1 * b_{abs,TR,950} + c_2 * b_{abs,WB,470} + c_3$          (17)

If CM is determined independently, $c_1$ and $c_2$ can be obtained by solving Equation 17, relating the light absorption to the particulate mass of both sources. The intercept $c_3$ represents a constant background concentration of non-absorbing carbonaceous material ($CM_{other}$). While Sandradewi et al. (2008a) did not require $CM_{other}$ to achieve mass closure, Favez et

al. (2010), Harrison et al. (2013) and Herich et al. (2011) found significant contributions of $CM_{other}$.

In practice, site-specific $c_1$, $c_2$ and $c_3$ values may be either fitted using equation 17 or less commonly fixed based on the knowledge of the OM-to-EC ratios in the primary emissions of interest (most frequently, only $c_1$ is fixed, e.g. Favez et al., 2010). The two approaches do not necessarily lead to the same result as they are not based on the same conceptual definitions of the organic aerosol fractions. When derived from the multiple linear regression fitting of equation 17, $CM_{TR}$

and $CM_{WB}$ would represent the fractions that correlate with $b_{abs,TR,950}$ and $b_{abs,WB,470}$, respectively. As SOA production is often very rapid (Huang et al., 2014), $CM_{TR}$ and $CM_{WB}$ are also expected to partially contain not only primary OA, but also SOA produced through the aging of traffic and wood burning emissions, respectively. Note that correlation is not causation and some of these correlations are a direct consequence of meteorology and planetary boundary layer mixing. This may significantly complicate data interpretation. Using this methodology, Herich et al. (2014) could not precisely quantify the

contributions of the different CM sources. They found a standard error of ±30% for $c_1$, $c_2$ and $c_3$ and a high sensitivity of $c_1$ and $c_2$ on the chosen $\alpha$ values for wood burning and traffic emissions. This is one of the few cases where errors related to CM apportionment using Aethalometer data are explicitly estimated. Usually only the sensitivity of $c_1$, $c_2$ and $c_3$ on the chosen $\alpha$ values are reported. However, the most profound flaw in the application of Eq. 17, using a multiple linear regression is the assumption that $c_3$, which is non-absorbing SOA mass mostly, is constant overtime. Consequently,

assessing how this mass is apportioned among the different sources and model residuals remain elusive and more faithful representation of the complex atmospheric processes would necessitate the use of a robust tracer for SOA. Accordingly, we do not recommend the use of this model in its current state to apportion CM mass, especially when the SOA fraction is dominant and highly variable.

A more conservative and controlled approach is to fix in the model the values of $c_1$ and $c_2$, based on emission data and

attribute the time-dependent remainder ($c_3$) to SOA. Under these conditions, $CM_{TR}$ and $CM_{WB}$ relate to the primary fraction and the model may better capture the time variability of SOA. Indeed, this approach would entail the precise knowledge of $c_1$





and $c_2$. OM-to-EC ratios in traffic emissions are heavily dependent on the type of fuel used, with lower values reported for diesel exhausts. Accordingly, for a European fleet dominated by diesel cars, El Haddad et al. (2013) report OM-to-EC ratios ranging between 0.25-0.45, whereas in the U.S., ratios between 0.9 and 1.4 were found (Zhang et al., 2005; Sun et al., 2012; Stroud et al., 2012). As biomass burning is a poorly controlled combustion process, typical OM-to-EC ratios determined at

emissions are highly scattered, ranging between 3 and 63 (Schauer et al., 2001; Fine et al., 2001, 2002, 2004a, 2004b; Schmidl et al., 2008). Nevertheless, more useful information may be obtained from examining ambient measurements, where, depending on the approach used to quantify wood smoke, OM-to-EC ratios may range between 3and 18 (Favez et al., 2010; Herich et al., 2014; Zotter et al., 2014).While these ratios must be selected with extreme caution and the sensitivity of the source apportionment results to this selection must be systematically assessed, additional on-site data (e.g. levoglucosan,

$^{14}$C…) may always aid constraining their values.

## 4. Conclusions

In this study, we show a comparison of the source apportionment of black carbon (BC) using the Aethalometer model with radiocarbon ($^{14}$C) measurements of elemental carbon (EC). This enables a validation of the choice of the Ångstrom exponents for wood burning ($\alpha_{WB}$) and traffic ($\alpha_{TR}$) emissions which have to be assumed a priori in the Aethalometer model.

Data from several campaigns across Switzerland with parallel Aethalometer and $^{14}$C measurements of the EC fraction from eight different stations with different characteristics allow the investigation of the applicability and performance of the Aethalometer model for different locations and conditions.

To obtain the best $\alpha$ pair in the Aethalometer model, outputs (using the 470 nm and 950 nm channels) were fitted against the fossil fraction of EC ($EC_F/EC$) derived from $^{14}$C measurements resulting in $\alpha_{TR} = 0.9$ and $\alpha_{WB} = 1.68$. Source apportionment

results from both methods, Aethalometer and $^{14}$C, are well correlated ($r = 0.81$) and the fitting residuals exhibit only a minor positive bias of 1.6% and an average precision of 9.3%, indicating that the Aethalometer model reproduces reasonably well the $^{14}$C results for all stations investigated in this study using our best estimate of a single $\alpha_{WB}$ and $\alpha_{TR}$ pair. We show that the Aethalometer model also works for other areas than for polluted alpine valleys in winter, since this analysis includes data from urban stations as well as days from spring and summer.

The residuals of the Aethalometer model outputs ($\Delta BC_{TR}/BC$) calculated with other $\alpha$ pairs depend on $EC_F/EC$ and a good agreement (within the $1\sigma$ confidence interval of $\Delta BC_{TR}/BC$) cannot be obtained over the entire $EC_F/EC$ range using other $\alpha$ pairs. However, combinations of $\alpha_{TR} = 0.8$, and 1.0 and $\alpha_{WB} = 1.8$ and 1.6, respectively, are also possible but only for a range of $EC_F/EC$ between ~40% and ~85%. The $\alpha_{WB}$ values previously used in Aethalometer (~2) models and any combination with $\alpha_{TR} = 1.1$ yield significant positive biases in the fitting residuals. Therefore we recommend to use the best $\alpha$

combination as obtained here ($\alpha_{TR} = 0.9$ and $\alpha_{WB} = 1.68$ for the wavelength pair 470 & 950 nm) in future studies. We also tested the sensitivity of the Aethalometer model due to different pairs of near UV (N-UV) and near IR (N-IR) wavelengths (470 & 950 nm, 470 & 880 nm and 370 & 880 nm). Any combination with 370 nm as N-UV wavelength resulted in larger residuals, a significant number of negative points and weaker correlations with $EC_F/EC$ and, therefore, we recommend to use 470 nm as N-UV wavelength in the Aethalometer model. Using 950 nm or 880 nm as N-IR wavelengths showed similar

results, though the former wavelength performed slightly better in this study.

Having an independent measurement for the relative contributions of traffic and wood-burning to total EC from the 14C and Aethalometer measurements, also made it possible to assess the dependence of the mass absorption cross section (MAC) of BC on its source. Results indicate no significant difference in MAC at 880nm (with a value of 11.8 $m^2g^{-1}$) between BC originating from traffic or wood burning emissions.

Applying the Aethalometer model to apportion total carbonaceous material (CM) it is evident that there are significant uncertainties and model errors (mainly due to assuming constant fitting parameters relating the absorption of traffic, wood-




burning and the residuals (comprising non-light absorbing CM, mostly secondary organic aerosol) to the separately determined CM). Therefore, in our opinion such a CM apportionment should only be interpreted qualitatively.

The results obtained in this study demonstrate that the evaluation of the choice of $\alpha_{WB}$ and $\alpha_{TR}$ using a reference method is highly valuable and should be performed when applying the Aethalometer model, if possible. In the absence of such reference measurements, however, assuming a single set of $\alpha_{TR}$ and $\alpha_{WB}$ yields acceptable results (i.e. average precision of 9.3% of $BC_{TR}/BC$ compared to $EC_F/EC$ in our case) and provides the best estimate of the fossil and non-fossil contributions to BC as apportioned by the Aethalometer model. Nevertheless these results were obtained for locations impacted by BC mainly from traffic consisting of a modern car fleet and wood combustion for residential heating in winter with well constrained combustion efficiencies. Furthermore, mainly winter conditions with only a few summer samples were available. Therefore, additional studies about the performance of the Aethalometer model with respect to seasonality and for sites with different combustion conditions and efficiencies, sources and fuels used and their temporal evolution are needed to reduce the uncertainties of their choice in the Aethalometer model.

**Acknowledgements**

This work was funded by the Swiss Federal Office for the Environment (BAFU), inNet Monitoring AG, OSTLUFT, the country Liechtenstein and the Swiss cantons Basel-Stadt, Basel-Landschaft, Graubünden, St. Gallen, Solothurn, Valais and Ticino. MG was supported by the ERC under grant ERC-CoG 615922-BLACARAT.

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




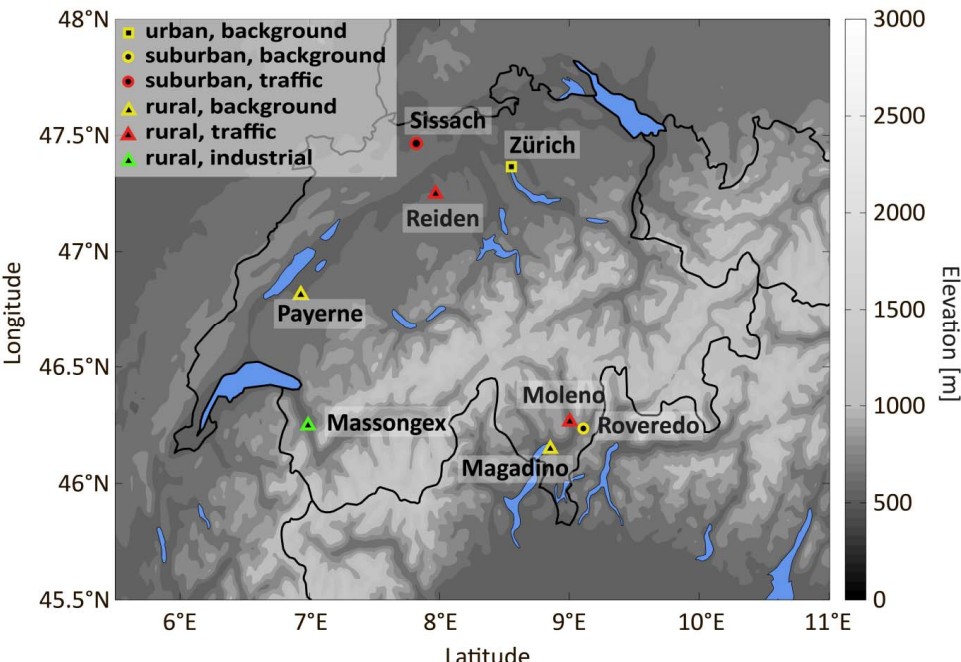

**Figure 1: Location of the different stations in Switzerland investigated in this study.**

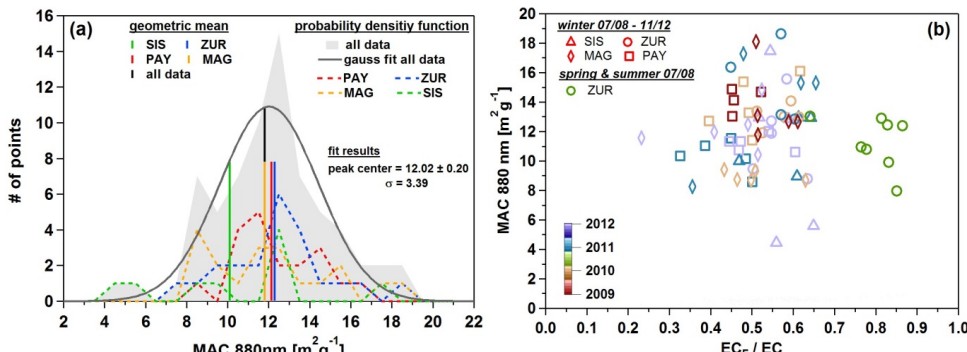

Figure 2: Distribution of MAC values of BC at 880 nm (a) and comparison with the fossil fraction of EC ($EC_F$/EC)
determined with the [14]C analysis (b). MAC values were determined assuming *C* value of 2.14 for the Aethalometer
and the EUSAAR-2 thermal optical transmission protocol was used for EC mass measurement. Only data from the
[14]C project Switzerland are included, since in earlier campaigns EC concentrations were not determined or measured
with the same TOT protocol.





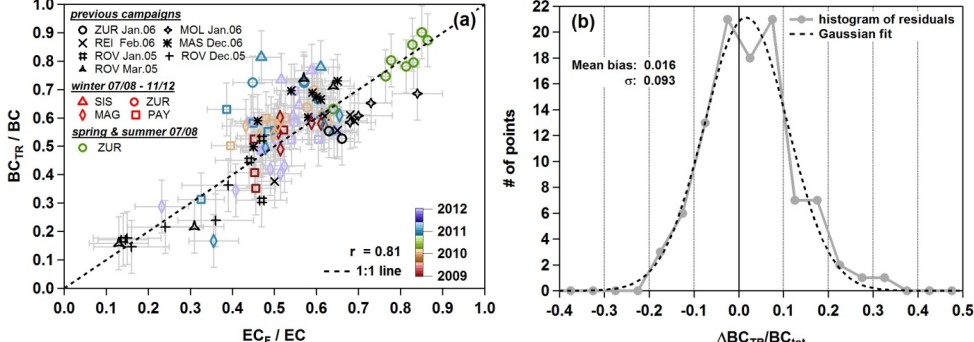

**Figure 3: (a)** Comparison between $BC_{TR}/BC$ at 950 nm and $EC_F/EC$ and **(b)** residuals of $BC_{TR}/BC$ compared to $EC_F/EC$ ($\Delta BC_{TR}/BC$). $BC_{TR}/BC$ was calculated using $b_{abs,470}$, $b_{abs,950}$, $MAC_{TR}/MAC_{WB} = 1$ and the $\alpha$ values ($\alpha_{WB} = 1.68$ and $\alpha_{TR} = 0.90$) obtained by fitting Equation 13 against $EC_F/EC$. The error bars for $EC_F/EC$ represent the combined uncertainty of the [14]C measurement of EC, the extrapolation of $f_{M,EC}$ to 100% EC yield and the bomb peak correction (see Sect. 2.3). The error bars for $BC_{TR}/BC$ denote the standard deviation ($\sigma$) of the Gaussian fit of $\Delta BC_{TR}/BC$ as obtained in Figure 3b.

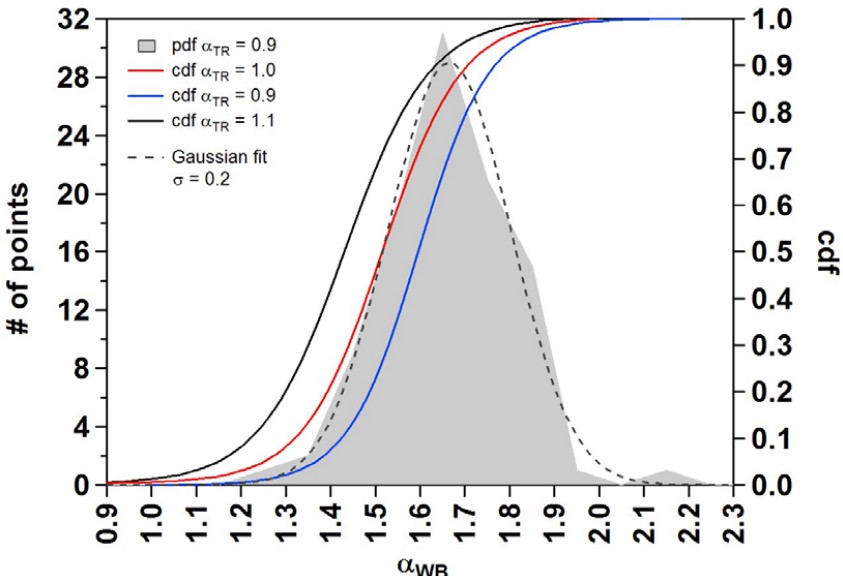

**Figure 4:** Histogram (pdf) and cumulative probability density function (cdf) of $\alpha_{WB}$ if $\alpha_{WB}$ is calculated for every data point, assuming a fixed $\alpha_{TR}$ (0.9, 1.0 or 1.1) and using $b_{abs,470}$, $b_{abs,950}$ and $MAC_{TR}/MAC_{WB} = 1$.





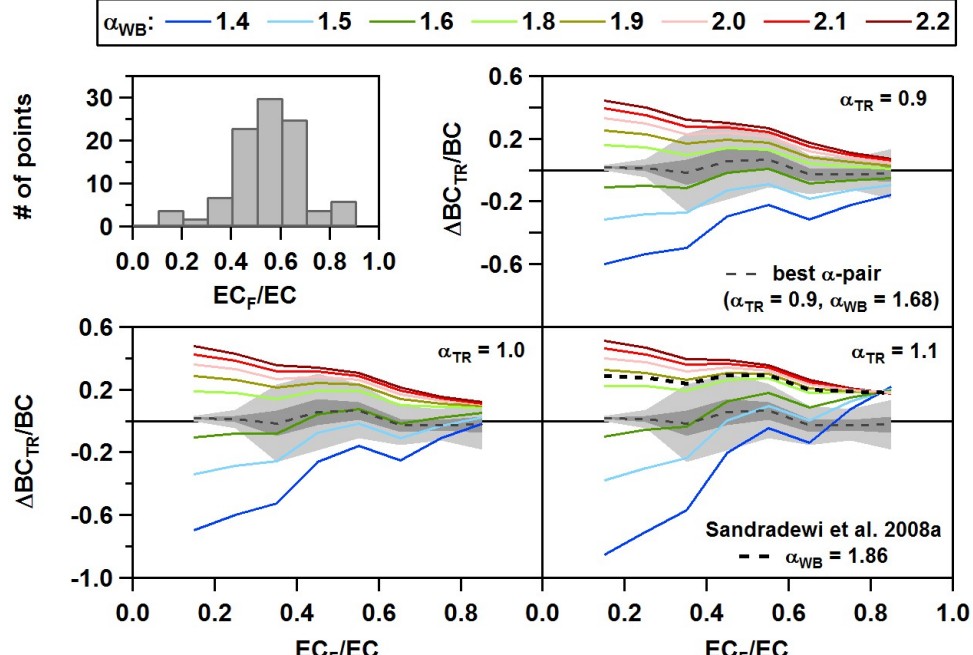

**Figure 5: Residuals of BC$_{TR}$/BC compared to EC$_F$/EC ($\Delta$BC$_{TR}$/BC) as a function of EC$_F$/EC for different combinations of $\alpha_{TR}$ and $\alpha_{WB}$. Average $\Delta$BC$_{TR}$/BC values for EC$_F$/EC bins of 0.1 are calculated for $\alpha_{WB}$ = 1.4-2.2 and a) $\alpha_{TR}$ = 0.9 (a), $\alpha_{TR}$ = 1.0 (b) and $\alpha_{TR}$ = 1.1 (c). $\Delta$BC$_{TR}$/BC for $\alpha_{TR}$ = 0.8 can be found in Figure S2 in the supporting information. The number of points per EC$_F$/EC bin is displayed in panel (a). The dashed grey line denotes the best $\alpha$ pair ($\alpha_{TR}$ = 0.9 and $\alpha_{WB}$ = 1.68) as obtained in Sect. 3.2.1 and the dark and light grey shaded areas mark the 1$\sigma$ (standard deviation) and 3$\sigma$ of $\Delta$BC$_{TR}$/BC per EC$_F$/EC bin for this best $\alpha$ pair. The black dashed line in (c) represents the $\alpha$ combination obtained by Sandradewi et al. (2008a) who first used the Aethalometer model.**

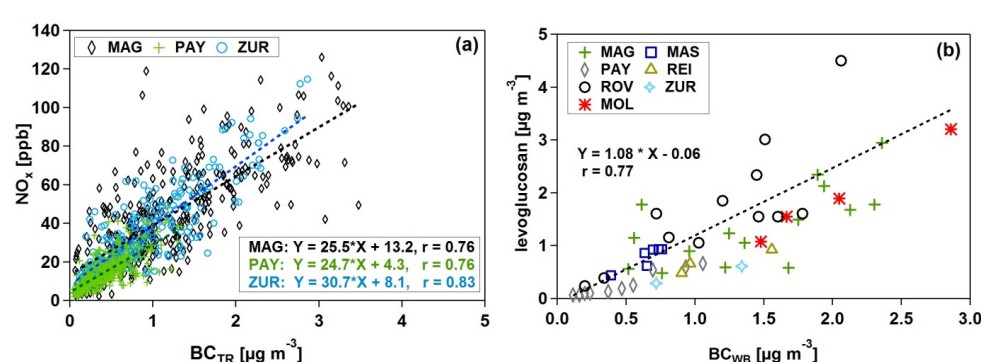

**Figure 6: Comparison of the Aethalometer model outputs calculated with the $\alpha$ values ($\alpha_{WB}$ = 1.68 and $\alpha_{TR}$ =0.90) obtained by fitting equation 13 against EC$_F$/EC with the additional traffic (NO$_x$) and wood burning (levoglucosan) markers: a) the correlation between BC$_{TR}$ at 950 nm and NO$_x$ averaged to 24 hours; and b) the scatterplot between the BC$_{WB}$ at 950 nm and levoglucosan. Details about the origin of the data are given in Table 1.**





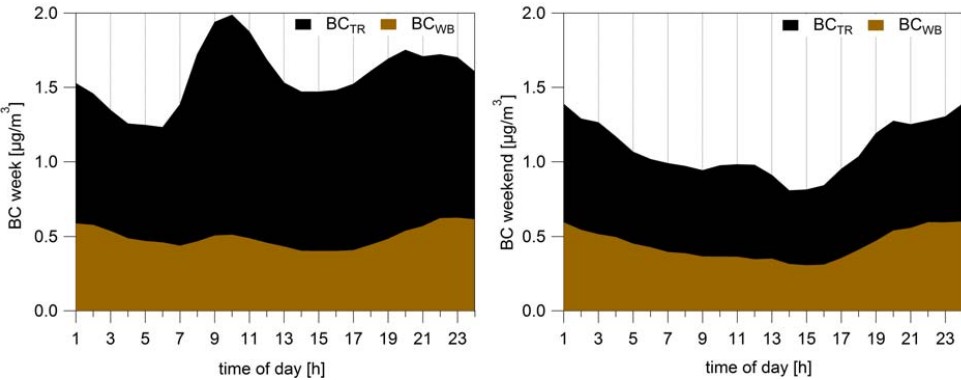

**Figure 7: Diurnal cycle at ZUR including 1h winter data from 2009 to 2012. BC$_{WB}$ and BC$_{TR}$ were calculated using the best $\alpha$ pair ($\alpha_{TR}$ = 0.9 and $\alpha_{WB}$ = 1.68) as obtained in Sect. 3.2.1. The split uncertainty between BC$_{WB}$ and BC$_{TR}$ ($\Delta$BC$_{TR}$/BC) is max. 0.04 µg m$^{-3}$.**

30




**Table 1: List of all stations and their classification according to the Swiss Federal Office for the Environment (BAFU) and additional campaign details.**

| sampling site | station code | station type | campaign | sampl. time | size cut filter/AETH | measurements | reference |
|---|---|---|---|---|---|---|---|
| Roveredo | ROV | suburban/ background | Jan. 2005 | 16h | $PM_{10}$/TSP | levoglucosan Aethalometer $^{14}C$ in EC | Szidat et al., 2007 Sandradewi et al., 2008a Sandradewi et al., 2008b Perron et al., 2010 |
| | | | Mar. 2005 | 16h | $PM_1$/$PM_{2.5}$ | | |
| | | | Dec. 2005 | 24h | $PM_1$/$PM_{2.5}$ | | |
| Moleno | MOL | rural/highway | Feb. 2005 | 16h | $PM_{10}$/$PM_{10}$ | | |
| Reiden | REI | rural/highway | Jan. & Feb. 2005 | 24h | $PM_{10}$/$PM_1$ | | |
| Massongex | MAS | rural/ industrial | Nov. & Dec. 2005 | 24h | $PM_{10}$/$PM_1$ | | |
| Zürich | ZUR | urban/ background | Jan. 2006 | 17h/ 40h | $PM_1$/$PM_1$ | | |
| Zürich | ZUR | urban/ background | $^{14}C$ project Switzerland (winter 2007/2008– 2011/2012) | 24h | $PM_{10}$/ $PM_{2.5}$ | levoglucosan[*]Ae thalometer[**] $^{14}C$ in EC[#] $NO_x$[‡] | Zotter et al., 2014 Herich et al., 2011 Herich et al., 2014 and references therein |
| Magadino | MAG | rural/ background | | | | | |
| Payerne | PAY | rural/ background | | | | | |
| Sissach | SIS | suburban/ traffic | | | | | |

[*]Levoglucosan was measured for ZUR, MAG and PAY for the winter 2008 and 2009 during the $^{14}C$ project Switzerland (see
Zotter et al. (2014) for more details). Additional data from these 3 stations were taken from Herich et al. (2011).
[**]Aethalometer measurements are continuously performed at the NABEL stations MAG and PAY since 2008 and ZUR since
2009. Data from these stations until January 2011 were already published in Herich et al. (2011) and data from 2011 and
2012 were provided by the NABEL network. An Aethalometer was additionally placed in SIS during winter 2010/2011 and
2011/2012.
[#]$^{14}C$ results of EC from all stations are presented in Zotter et al. (2014)
[‡]$NO_x$ is continuously measured at the NABEL stations MAG, PAY and ZUR using reference instrumentation with
molybdenum converters according to valid European standards (see Herich et al. (2011), EMPA (2013) and Zotter et al.
(2014) for more details).



**Table 2: Ranges and averages of $\alpha_{WB}$ values resulting in a ratio of 1 between $BC_{TR}/BC$ (at 950 nm) and $EC_F/EC$ for all stations calculated with $\alpha_{TR}$ of 0.9, $b_{abs,470}$, $b_{abs,950}$ and $MAC_{TR}/MAC_{WB} = 1$.**

| station | $\alpha_{WB}$ range | $\alpha_{WB}$ mean ± standard deviation |
|---|---|---|
| SIS | 1.23–1.84 | 1.55 ± 0.21 (n = 9) |
| ZUR (winter) | 1.47–1.80 | 1.67 ± 0.11 (n = 14) |
| ZUR (summer) | 1.34–1.90 | 1.60 ± 0.14 (n = 8) |
| MAG | 1.53–1.85 | 1.69 ± 0.09 (n = 19) |
| PAY | 1.42–1.80 | 1.63 ± 0.10 (n = 19) |
| MOL | 1.85–2.17 | 1.93 ± 0.16 (n = 4) |
| ROV | 1.43–1.85 | 1.68 ± 0.11 (n = 13) |
| REI | 1.70–1.86 | 1.81 ± 0.06 (n = 5) |
| MAS | 1.46–1.65 | 1.56 ± 0.06 (n = 8) |

**Table 3: Evaluation of the Aethalometer model using different wavelength pairs. a) calculation of the $\alpha$ values by fitting Equation 13 ($MAC_{TR}/MAC_{WB} = 1$) against $EC_F/EC$. b) comparison between $EC_F/EC$ and $BC_{TR}/BC$ calculated with $\alpha_{WB} = 1.68$ and $\alpha_{WB} = 0.90$, representing the best $\alpha$ pair for all data, for different wavelength pairs. $\mu$ and $\sigma$**
10 **denote the center and the standard deviation of the Gaussian fit of the residuals of $BC_{TR}/BC$ compared to $EC_F/EC$, respectively.**

| a) | Calculation of best α values | | | |
|---|---|---|---|---|
| wavelength pair | $\alpha_{WB}$ | $\alpha_{TR}$ | $\mu$ of $\Delta BC_{TR}/BC$ | $\sigma$ of $\Delta BC_{TR}/BC$ |
| 470 & 950 nm | 1.68 | 0.90 | 2% | 9% |
| 470 & 880 nm | 1.75 | 0.90[*] | 7% | 11% |
| 370 & 950 nm | 2.09 | 0.90[*] | 17% | 12% |
| 370 & 880 nm | 2.09 | 0.90[*] | 18% | 13% |

| b) | $EC_F/EC$ vs. $BC_{TR}/BC$ with $\alpha_{TR} = 0.90$ and $\alpha_{WB} = 1.68$ | | |
|---|---|---|---|
| wavelength pair | mean $\Delta BC_{TR}/BC$ | negative $BC_{TR}/BC$ points | $r$ (with $EC_F/EC$) |
| 470 & 950 nm | 2% | 0% | 0.80 |
| 470 & 880 nm | 3% | 3% | 0.65 |
| 370 & 950 nm | -12% | 16% | 0.63 |
| 370 & 880 nm | -15% | 19% | 0.58 |

[*]no physically meaningful value for $\alpha_{TR}$ could be obtained by the fitting Equation 13 against $EC_F/EC$ and therefore, $\alpha_{TR}$ was set to 0.9 representing the best value for the wavelength pair 470 nm and 950 nm.

