# Peer review of "Evaluation of the absorption Ångström exponents for traffic and wood burning in the Aethalometer based source apportionment using radiocarbon measurements of ambient aerosol"

_Atmospheric Chemistry and Physics, 2016_

## Referee Comment (RC1) · Anonymous Referee #4 · 10 Nov 2016

Summary: The analysis outlined in this manuscript utilizes concurrent Aethalometer and 14C measurements from various locations throughout Switzerland to (1) determine absorption Ångström exponents for BC from vehicle emissions and from wood burning emissions, (2) assess Aethalometer measurements and absorption Ångström exponent values in different geographical contexts, seasons, and conditions, and (3) Assess MAC values for different BC sources.

General Comments: Very well written, I found very few technical corrections. Authors should expand on discussion of uncertainties and implications associated with

choice of Aethalometer correction algorithm, with particular emphasis on how corrections can affect absorption Ångström exponent. Throughout manuscript: For consistency in the field, please consider using the terminology 'Equivalent Black Carbon', or EBC, to describe measurements of BC from optical absorption methods (in this case, the Aethalometer), as described in Petzold et al. (2013. From Petzold et al. (2013): "Equivalent black carbon (EBC) should be used instead of black carbon for data derived from optical absorption methods, together with a suitable MAC for the conversion of light absorption coefficient into mass concentration."

Specific Comments: Page 5, Line 4-6: The literature suggests that absorption Ångström exponent is affected by choice of correction algorithm, though these impacts are not well outlined (Collaud Coen et al., 2010). Parts of the correction schemes that contribute to wavelength dependence include: (1) filter loading artefact, (2) discrepancy between scattering and backscattering effect, (3) effects of particles getting embedded in filter material. Since absorption Angstrom exponent is the main focus of this paper, it may be prudent to mention here that your results might be sensitive to your choice of correction algorithm, even if it's not clear exactly how. This is future work that needs to be done. Page 5, Line 27-29: Citations needs for BC and traffic absorption Ångström exponent numbers, and for claim that traffic emissions contain mainly BC. Page 5, Line 36-38: Great that you are addressing this assumption in the context of the specific geographical area in which the measurements are taken. Are emissions from dust, light absorbing SOA, biogenic emissions also negligible in this region? Maybe there is a source to cite here to address the validity of this assumption, or maybe not. Page 9, Lines 11-29: Shouldn't you compare your MAC value here to the MAC values that are automatically programmed and preset in the Aethalometers? They are available in the AE31 manual. Page 14, Line 15: Nowhere is it specified or explained what OM is in Equation 16. It presumably organic matter, but should be stated explicitly in the text for readers unfamiliar. Figure 1 & Table 1: Would be useful to state in the text how station types were determined; for example, what is the difference between urban, background vs. urban, traffic, etc.

Technical Corrections: Page 13, Line 4-5: There are a few commas missing that make this sentence difficult to read. The sentence should read: "However, as shown here, the choice of the wavelengths, especially the one in the N-UV range, and $\alpha$WB are not independent". Page 14, Line 13: "apportioned" should be "apportion" so the sentence reads: "It has been attempted to also apportion the total carbonaceous. . ." Page 15, Line 13: should specify absorption Angstrom exponent.

References:

Collaud Coen, Martine, et al. "Minimizing light absorption measurement artifacts of the Aethalometer: evaluation of five correction algorithms." Atmospheric Measurement Techniques 3 (2010): 457-474.

Petzold, Andreas, et al. "Recommendations for reporting" black carbon" measurements." Atmospheric Chemistry and Physics 13.16 (2013): 8365-8379.

---

## Referee Comment (RC2) · Anonymous Referee #5 · 6 Dec 2016

the authors take the advantage of a multi-year, multi-campaign dataset in order to investigate a very important technical issue in the context of source apportionment. They do manage to make their point in a very well structured paper. I believe this is an important paper for the future of the field.

---

## Author Comment (AC1) · 30 Jan 2017

We thank the reviewer for the positive comments and are very pleased that our manuscript left the reviewer satisfied without further remarks.

---

## Author Comment (AC2) · 30 Jan 2017

**Responses to Reviewer's Comments to**

Zotter et al., Atmos. Chem. Phys. Discuss., 2016: p. 1-29. *"Evaluation of the absorption Ångström exponents for traffic and wood burning in the Aethalometer based source apportionment using radiocarbon measurements of ambient aerosol"*

We thank the reviewer for his comments on our paper. To guide the review process we have copied the reviewers' comments in italic; and our responses are in blue, regular font. We have responded to all the referee comments and done the modifications accordingly (**in bold in the text**).

**Anonymous Referee #4**

*R4.1 Summary:*
*The analysis outlined in this manuscript utilizes concurrent Aethalometer and 14C measurements from various locations throughout Switzerland to (1) determine absorption Ångström exponents for BC from vehicle emissions and from wood burning emissions, (2) assess Aethalometer measurements and absorption Ångström exponent values in different geographical contexts, seasons, and conditions, and (3) Assess MAC values for different BC sources.*

We thank the reviewer for the positive comments. In the following we will respond to each comment listed below separately.

*General Comments:*

*R4.2: Very well written, I found very few technical corrections. Authors should expand on discussion of uncertainties and implications associated with choice of Aethalometer correction algorithm, with particular emphasis on how corrections can affect absorption Ångström exponent.*

The results of our study for $\alpha_{WB}$ and $\alpha_{TR}$ were obtained from equation 13 and 14. In these equations only fractional contributions of $BC_{TR}$ to total BC, and not absolute values, are considered. Therefore, Aethalometer compensation algorithms, i.e. for filter loading and scattering effects, which especially affect the absolute eBC concentrations, were not discussed in details and can be found in Collaud Coen et al. (2010). We have only investigated in Figure 2 the stability of the $MAC_{BC}$ values, which relate the absorption measurements to the BC mass. The variability in these values is ~25%, and this includes uncertainties in both aethalometer and sunset measurements, indicating that the compensation algorithms are robust for the absorption measurements at N-IR under our conditions. While systematic biases in the compensation factors do not influence the determination of $\alpha_{WB}$ and $\alpha_{TR}$, errors in the treatment of the wavelength dependence of the compensation factors by the different algorithms available may have an influence. Despite this, biases in the non-compensated $b_{abs}$ due to the filter loading effect are rather small compared to the scattering compensation (f-values used are 1.051-1.155 vs. C = 2.14). Hence the uncertainties in the obtained $\alpha_{WB}$ and $\alpha_{TR}$ due to wavelength dependence of the filter loading effect compensation parameters from different compensation algorithms will be small. In addition, depending on the compensation algorithm used, by carefully checking the data the performance of the filter loading compensation can be assessed which further eliminates uncertainties. This is already mentioned in the original manuscript (see below or ACPD version of manuscript page 6, lines 24-26).

"Nevertheless, uncertainties in the $BC_{TR}$ to BC ratio associated with the filter loading compensation can be kept small by carefully determining the *f* values, following the approach in Weingartner et al. (2003) or Sandradewi et al. (2008c). The Aethalometer AE33 measures the compensation parameters

and therefore the compensation is performed on-line. The precision of this compensation can be checked using the BC(ATN) or $b_{abs}$(ATN) analysis (Drinovec et al., 2015)."

Uncertainties in the obtained $\alpha_{WB}$ and $\alpha_{TR}$ due to different wavelength dependencies of the scattering compensation in different compensation algorithms can introduce additional uncertainties. To quantify the possible influence of using different compensation algorithm, data presented would have to be compensated with different compensation methods and the best $\alpha_{TR}$ and $\alpha_{WB}$ combination as well as the fitting residuals have to be determined for every method. However, for almost all of the available Aethalometer compensation algorithms concurrent scattering measurements are necessary, which are not available for the dataset presented in this study. A detailed comparison of the different compensation methods is available in Collaud Coen et al. (2010), and is beyond the scope of this work, although this influence has been already considered in the ACPD version of the manuscript, by the statement on page 6, lines 16-22, which reads as follows:

"Additional uncertainties may arise from the compensation factors applied to the attenuation coefficients. In this study, a fixed $C_\lambda$ value was used for the multi-scattering correction (Sect. 2.2.1) and thus the ratio $C_{\lambda,1}/C_{\lambda,2}$ becomes unity in Eq. 14. This is justified and introduces very little uncertainty, as the wavelength dependence of the $f$ and $C$ values across the range $\lambda = 470$-950 nm was reported to be less than 10% and 12%, respectively, for the Aethalometer model AE31 (Weingartner et al., 2003; Sandradewi et al., 2008c; Segura et al., 2014). If data from other photometer models, which exhibit a wavelength dependence of the C value, are used for the source apportionment, the correct ratio $C_{\lambda,1}/C_{\lambda,2}$ must be used in Eq. 14 to ensure consistency of the Aethalometer model parameters."

In addition, we do provide an overall assessment of the uncertainties by examining the fitting residuals between $BC_{TR}$/BC and $EC_F$/EC assuming the fitted $\alpha_{WB}$ and $\alpha_{TR}$ combinations. As mentioned in the manuscript (ACPD manuscript page 10, lines 27-30), these residuals are driven by (1) random measurement uncertainties of $EC_F$/EC, and $b_{abs,470}$ and $b_{abs,950}$ and (2) day-to-day and station-to-station variability (see Table 2 and Figure 3 in the manuscript). The measurement uncertainty in $EC_F$/EC is estimated to be around 5-6%, while the average total uncertainty is estimated to be around ~15% (Figure 5). Therefore, it is not expected that uncertainties in $b_{abs,470}$ and $b_{abs,950}$ due to different Aethalometer data compensation algorithms will additionally have a significant effect on the determination of the best $\alpha_{TR}$ and $\alpha_{WB}$ combination.

Based on the reviewer comment we have added the following discussion:

Revised manuscript, page 7 lines 2-10:
**"It should be noted that the calculation of the $EBC_{TR}$ to EBC ratio (Equation 13) might not only be sensitive to the choice of compensation parameters but also on the choice of compensation algorithm. However, large uncertainties of the $EBC_{TR}$ to EBC ratio due to the use of different Aethalometer data compensation algorithms are not expected since in Eq. 13 and Eq. 14 only fractional contributions of $b_{abs}(\lambda)$ or $b_{ATN}(\lambda)$ are used. Therefore, only differences in the wavelength dependency of the compensation parameters in different compensation methods would slightly affect the determination of $EBC_{TR}$/EBC. An investigation of such effects is beyond the scope of this study, however, future work should be carried out exploring possible influences of different compensation methodologies on $EBC_{TR}$/EBC. A detailed comparison of the different Aethalometer compensation algorithms can be found in Collaud Coen et al. (2010) and only an overall assessment of the methodology used will be discussed below."**

*R4.3: Throughout manuscript: For consistency in the field, please consider using the terminology 'Equivalent Black Carbon', or EBC, to describe measurements of BC from optical absorption methods (in this case, the Aethalometer), as described in Petzold et al. (2013. From Petzold et al. (2013): "Equivalent black carbon (EBC) should be used instead of black carbon for data derived from optical absorption methods, together with a suitable MAC for the conversion of light absorption coefficient into mass concentration."*

Throughout the revised manuscript (in the main text, the SI, figures and tables), BC was replaced by EBC when BC was used in context with the Aethalometer. In addition, to explain the use of EBC the following text was added on page 2, lines 13-14:

"The quantities measured are defined based on the instrument and protocol used, with BC and EC related to  **optical** and thermo-optical **as well as chemical** measurements, respectively**. When BC is obtained by light absorption measurements it is referred to as mass equivalent black carbon (EBC)** (Petzold et al., 2013)."

*Specific Comments:*

*R4.4: Page 5, Line 4-6: The literature suggests that absorption Ångström exponent is affected by choice of correction algorithm, though these impacts are not well outlined (Collaud Coen et al., 2010). Parts of the correction schemes that contribute to wavelength dependence include: (1) filter loading artefact, (2) discrepancy between scattering and backscattering effect, (3) effects of particles getting embedded in filter material. Since absorption Angstrom exponent is the main focus of this paper, it may be prudent to mention here that your results might be sensitive to your choice of correction algorithm, even if it's not clear exactly how. This is future work that needs to be done.*

We agree with the reviewer and add the following remarks in the revised manuscript:

**"It should be noted that these different compensation algorithms might yield slightly different $b_{abs}(\lambda)$."** (revised manuscript page 5, lines 6-7)

**"It should be noted that the calculation of the $EBC_{TR}$ to EBC ratio (Equation 13) might not only be sensitive to the choice of compensation parameters but also on the choice of compensation algorithm. However, large uncertainties of the $EBC_{TR}$ to EBC ratio due to the use of different Aethalometer data compensation algorithms are not expected since in Eq. 13 and Eq. 14 only fractional contributions of $b_{abs}(\lambda)$ or $b_{ATN}(\lambda)$ are used. Therefore, only differences in the wavelength dependency of the compensation parameters in different compensation methods would slightly affect the determination of $EBC_{TR}/EBC$. An investigation of such effects is beyond the scope of this study, however, future work should be carried out exploring possible influences of different compensation methodologies on $EBC_{TR}/EBC$. A detailed comparison of the different Aethalometer correction algorithms can be found in Collaud Coen et al. (2010) and only an overall assessment of the methodology used will be discussed below."** (Revised manuscript, page 6 lines 31 and page 7 lines 1-8, see also response to comment R4.2)

*R4.5: Page 5, Line 27-29: Citations needs for BC and traffic absorption Ångström exponent numbers, and for claim that traffic emissions contain mainly BC.*

The following four citations were inserted (see also revised manuscript page 5, lines 29-30): **Bond et al., 2013;Kirchstetter et al., 2004;Schnaiter et al., 2003;Schnaiter et al., 2005**

*R4.6: Page 5, Line 36-38: Great that you are addressing this assumption in the context of the specific geographical area in which the measurements are taken. Are emissions from dust, light absorbing SOA, biogenic emissions also negligible in this region? Maybe there is a source to cite here to address the validity of this assumption, or maybe not.*

We agree with the reviewer that citations for the assumptions of the Aethalometer model (only traffic and wood burning contribute to the absorption) should be listed. Since there is no single reference available the original statement on page 5, lines 36-38 was extended and explanations and citations, why other possible sources contribute to the absorption, can be neglected were added (see below and revised manuscript page 5, lines 37-39 and page 6, lines 1-4).

"This assumption is valid for Switzerland and other  **Alpine regions** in Europe, **especially in winter,** where emissions from other sources  are negligible. **Coal burning is not**

**used in these areas (Eurostat, 2017) and biogenic SOA is mostly absorbing in the UV range (Romonosky et al., 2016) not covered by wavelengths used in the Aethalometer (especially that we recommend the use of the absorption at 470nm rather than at 370nm). Mineral dust can usually be neglected in this region (contribution to total PM < 10% (Gianini et al., 2012)), and special events possibly influencing the absorption at Aethalometer wavelengths 470-590 nm can be identified due to a drop of the absorption Ångström exponent clearly below one during such events (Collaud Coen et al., 2004)."**

Contributions from light absorbing biomass burning SOA cannot be completely excluded in the Aethalometer model. This was already considered in the ACPD version with the following statement on page 10, lines 9-12 which explicitly includes that $\alpha_{WB}$ can depend on aerosol aging:

"... $\alpha_{WB}$ and $\alpha_{TR}$ may be highly variable, depending on the combustion conditions and efficiency, fuel type and aerosol aging (Lack et al., 2013; Saleh et al., 2013; Zhong and Jang, 2014; Sharpless et al., 2014; Saleh et al., 2014; Kirchstetter et al., 2004; Bond and Bergstrom, 2006 and references therein; Herich et al., 2011; Garg et al., 2016)."

We note though that while the $\alpha_{WB}$ might be higher than those of primary emissions influenced by the presence of light absorbing biomass burning organic aerosol (BBOA) SOA, the analysis that we propose for the quantification of the fraction of EBCTR to EBC ratio is not influenced. Indeed, different extent of aging would add to the variability of $\alpha_{WB}$, but as shown in Figure 5, the overall $\alpha_{WB}$ variability is ~15%. Therefore, we do not expect that the presence of SOA affects significantly the proposed model.

In addition, in section 3.2.4 of the ACPD version we also point out that lower wavelengths in the Aethalometer could be influenced by light absorbing non BBOA SOA, other absorbing non-BC combustion particles and atmospheric processing. Consequently, we recommended to rather use 470 nm, and not 370 nm, as the lower wavelength since higher wavelengths are expected to be less affected by other absorbing non-BC sources (see below and ACPD manuscript page 13, lines 5-11):

"Since 1) it was previously shown that adsorption of VOCs on the filter tape of the Aethalometer can occur which possibly influences the absorption measurement with the 370 nm channel (Vecchi et al., 2014), 2) light absorbing SOA, other absorbing non-BC combustion particles and atmospheric processing affects lower wavelengths more than higher ones and 3) our results indicate an inferior agreement of $BC_{TR}$/BC with ECF/EC using 370 nm as N-UV wavelength we therefore recommend using 470 nm as the N-UV wavelength in the Aethalometer model while the choice between 950 nm and 880 nm in the N-IR is less critical."

*R4.7: Page 9, Lines 11-29: Shouldn't you compare your MAC value here to the MAC values that are automatically programmed and preset in the Aethalometers? They are available in the AE31 manual.*

In the Aethalometer algorithm the attenuation is directly converted to a BC mass using an attenuation cross section (Hansen, 2005), presented in the manual. However, the attenuation is influenced by multiple scattering of the filter fibers and the loading effect and is, therefore, different to the absorption coefficient $b_{abs}$. The mass absorption cross section (MAC value) relates absorption, and not attenuation, with mass. Therefore, on page 9, lines 11-29 we do not compare our empirically derived MAC values with the values of the attenuation cross section provided in the Aethalometer model. Since in section 2.2.1 the differences between attenuation and absorption coefficient are explained in detail, we do not find it necessary to add further explanations in this section.

*R4.8: Page 14, Line 15: Nowhere is it specified or explained what OM is in Equation 16. It presumably organic matter, but should be stated explicitly in the text for readers unfamiliar.*

To specify OM the explanation of this abbreviation was added next to the equation (see below and revised manuscript page 14, line 29).

"$CM = OM + BC$,                   *OM…organic matter*                                        (16)"

*R4.9: Figure 1 & Table 1: Would be useful to state in the text how station types were determined; for example, what is the difference between urban, background vs. urban, traffic, etc.*

The terms "urban", "suburban" and "rural" are determined by density of buildings around the measurement station and the location away (several hundred meters or more) or within larger villages and cities. The terms "background", "industry" and "traffic"/"highway" describe the exposure to emission sources. The station types where defined by the Swiss Federal Office for the Environment and these terms are also used in many other comparable publications. We added a short description of these terms in the footnote of the table (see below and Table 1 in the revised manuscript on page 28, lines 15-22)

**urban:** **station is located within a larger village or city and is surrounded by buildings with a high building density**
**suburban:** **building density in the immediate surrounding of the station is low and there is only few traffic in the area**
**rural:** **hardly any buildings in surrounding of station, larger streets and village/city several hundred meters or more away**
**traffic:** **station is located directly at a street with considerable amount of traffic**
**highway:** **station is located next to a highway**
**industrial:** **station is located in an industrial area**
**background:** **no large influence of direct emissions from sources in near vicinity (e.g., traffic, industry or domestic)**

*Technical Corrections:*

*R4.10: Page 13, Line 4-5: There are a few commas missing that make this sentence difficult to read. The sentence should read: "However, as shown here, the choice of the wavelengths, especially the one in the N-UV range, and _WB are not independent".*

The commas were added accordingly (see below and revised manuscript page 13, lines 18-19).

"However, as shown here**,** the choice of the wavelengths, especially the one in the N-UV range**,** and $\alpha_{WB}$ are not independent."

*R4.11: Page 14, Line 13: "apportioned" should be "apportion" so the sentence reads: "It has been attempted to also apportion the total carbonaceous: : :"*

This was corrected accordingly (see below and revised manuscript page 14, line 27)

"It has been attempted to also apportion the total carbonaceous…"

*R4.12: Page 15, Line 13: should specify absorption Angstrom exponent.*

"Absorption" was inserted before "Angstrom" accordingly (see below and revised manuscript page 15 line 29).

"…choice of the **absorption** Ångström exponents for…"

**References:**

Collaud Coen, Martine, et al. "Minimizing light absorption measurement artifacts of the Aethalometer: evaluation of five correction algorithms." Atmospheric Measurement Techniques 3 (2010): 457-474.

Petzold, Andreas, et al. "Recommendations for reporting" black carbon" measurements." Atmospheric Chemistry and Physics 13.16 (2013): 8365-8379.

Bond, T. C., Doherty, S. J., Fahey, D. W., Forster, P. M., Berntsen, T., DeAngelo, B. J., Flanner, M. G., Ghan, S., Kärcher, B., Koch, D., Kinne, S., Kondo, Y., Quinn, P. K., Sarofim, M. C., Schultz, M. G., Schulz, M., Venkataraman, C., Zhang, H., Zhang, S., Bellouin, N., Guttikunda, S. K., Hopke, P. K., Jacobson, M. Z., Kaiser, J. W., Klimont, Z., Lohmann, U., Schwarz, J. P., Shindell, D., Storelvmo, T., Warren, S. G., and Zender, C. S.: Bounding the role of black carbon in the climate system: A scientific assessment, J. Geophys. Res.-Atmos., 118, 5380-5552, doi: 10.1002/jgrd.50171, 2013.

Collaud Coen, M., Weingartner, E., Schaub, D., Hueglin, C., Corrigan, C., Henning, S., Schwikowski, M., and Baltensperger, U.: Saharan dust events at the Jungfraujoch: detection by wavelength dependence of the single scattering albedo and first climatology analysis, Atmos. Chem. Phys., 4, 2465-2480, doi: 10.5194/acp-4-2465-2004, 2004.

Collaud Coen, M., Weingartner, E., Apituley, A., Ceburnis, D., Fierz-Schmidhauser, R., Flentje, H., Henzing, J. S., Jennings, S. G., Moerman, M., Petzold, A., Schmid, O., and Baltensperger, U.: Minimizing light absorption measurement artifacts of the Aethalometer: Evaluation of five correction algorithms, Atmos. Meas. Tech., 3, 457-474, doi: 10.5194/amt-3-457-2010, 2010.

Energy Statistics - Main Tables: http://ec.europa.eu/eurostat/web/energy/data/main-tables, access: 2017/01/11, 2017.

Gianini, M. F. D., Gehrig, R., Fischer, A., Ulrich, A., Wichser, A., and Hueglin, C.: Chemical composition of PM10 in Switzerland: An analysis for 2008/2009 and changes since 1998/1999, Atmos. Environ., 54, 97-106, doi: 10.1016/j.atmosenv.2012.02.037, 2012.

Hansen, A. D. A.: The Aethalometer, manual, Magee Scientific, Berkeley, California, USA, 2005.

Kirchstetter, T. W., Novakov, T., and Hobbs, P. V.: Evidence that the spectral dependence of light absorption by aerosols is affected by organic carbon, J. Geophys. Res.-Atmos., 109, D21208, doi: 10.1029/2004JD004999, 2004.

Romonosky, D. E., Ali, N. N., Saiduddin, M. N., Wu, M., Lee, H. J., Aiona, P. K., and Nizkorodov, S. A.: Effective absorption cross sections and photolysis rates of anthropogenic and biogenic secondary organic aerosols, Atmos. Environ., 130, 172-179, doi: http://dx.doi.org/10.1016/j.atmosenv.2015.10.019, 2016.

Sandradewi, J., Prevot, A. S. H., Weingartner, E., Schmidhauser, R., Gysel, M., and Baltensperger, U.: A study of wood burning and traffic aerosols in an Alpine valley using a multi-wavelength Aethalometer, Atmos. Environ., 42, 101-112, doi: 10.1016/j.atmosenv.2007.09.034, 2008c.

Schnaiter, M., Horvath, H., Möhler, O., Naumann, K. H., Saathoff, H., and Schöck, O. W.: UV-VIS-NIR spectral optical properties of soot and soot-containing aerosols, J. Aerosol. Sci., 34, 1421-1444, doi: 10.1016/S0021-8502(03)00361-6, 2003.

Schnaiter, M., Linke, C., Möhler, O., Naumann, K. H., Saathoff, H., Wagner, R., Schurath, U., and Wehner, B.: Absorption amplification of black carbon internally mixed with secondary organic aerosol, J. Geophys. Res.-Atmos., 110, D19204, doi: 10.1029/2005JD006046, 2005.

Segura, S., Estellés, V., Titos, G., Lyamani, H., Utrillas, M. P., Zotter, P., Prévôt, A. S. H., Močnik, G., Alados-Arboledas, L., and Martínez-Lozano, J. A.: Determination and analysis of in situ spectral aerosol optical properties by a multi-instrumental approach, Atmos. Meas. Tech., 7, 2373-2387, doi: 10.5194/amt-7-2373-2014, 2014.

Weingartner, E., Saathoff, H., Schnaiter, M., Streit, N., Bitnar, B., and Baltensperger, U.: Absorption of light by soot particles: determination of the absorption coefficient by means of aethalometers, J. Aerosol. Sci., 34, 1445-1463, doi: 10.1016/s0021-8502(03)00359-8, 2003.

---

## Author Response (AR2)

**Responses to Reviewer's Comments to**

Zotter et al., Atmos. Chem. Phys. Discuss., 2016: p. 1-29. *"Evaluation of the absorption Ångström exponents for traffic and wood burning in the Aethalometer based source apportionment using radiocarbon measurements of ambient aerosol"*

We thank the Co-Editor for the positive comments on our paper and pointing out that some minor corrections are needed.

Since we agree with all of the Co-Editors comments and there were only minor changes/corrections/typos we do not provide a point-by-point response of the comments. Instead all of the Co-Editor comments are accepted and incorporated in the "track changes mode" in the revised manuscript.

[revised manuscript text omitted]